# Transcriptome-wide mapping of small ribosomal subunits elucidates scanning mechanisms of translation initiation in the mammalian brain

Preeti Madhav Kute [1,2,6], Francois P. Pauzin[3,4,6], Kornel Labun[1,6], Clive R. Bramham [3,4] ✉ & Eivind Valen [1,2,5] ✉

Neuronal protein synthesis is highly compartmentalised and regulated, with key roles for translation initiation and elongation factors. Ribosome profiling, the most widely used transcriptome-wide method for measuring translation, captures translation elongation, but not the initiation phase involving small ribosomal subunit (SSU) scanning. Here, we adapted ribosome complex profiling (RCP-seq) for mouse dentate gyrus and cerebral cortex, to characterize translation initiation. In both tissues, SSUs accumulate near the start codon on synaptically localised RNAs, and this 'poised' SSU configuration correlates with enhanced translational efficiency. Upstream open reading frames (uORFs) are associated with less poised SSUs, potentially by disassociating the SSUs. We further find that neuron-specific transcripts recruit more ribosomes and are more efficiently translated than glia-specific transcripts. For neuronal transcripts, monosome-preferring mRNAs show less SSU occupancy relative to polysome-preferring mRNAs, suggesting reduced recruitment of ribosomes. In summary, RCP-seq elucidates translation initiation dynamics and cell-type- and transcript-specific regulation in the mammalian brain.

Eukaryotic translation can be divided into four distinct stages: initiation, elongation, termination, and ribosome recycling. However, knowledge of translation dynamics across these stages is limited due to the lack of methods with nucleotide resolution in live systems[1–4]. Translation initiation, a rate-limiting process in protein synthesis, can be further divided into recruitment, scanning, and start codon recognition, which can all be subject to regulation. During recruitment, the assembly of the preinitiation complex (PIC) containing the small ribosomal subunit (SSU/40S) with methionyl-initiator tRNA, and subsequent recruitment to the mRNA 5′ cap, is regulated by eukaryotic initiation factors (eIFs)[1]. Following recruitment, the PIC scans the 5′ leader for a start codon, after which the large ribosomal subunit (LSU/60S) joins the SSU to form an elongation-competent ribosome (80S).

Translation in neurons is highly compartmentalised and regulated due to their complex polar structure and function. A prominent example is the local translation in the synaptic compartment remote from the cell body, important for synapse development, maintenance, and plasticity[5–7]. Both translation initiation and elongation in the nervous system are modulated by changes in translation factor activity in response to changes in neural activity and neurotransmitter receptor activation. Converging evidence highlights the role of translation initiation factors (eIF2α, eIF4E) as regulators of protein synthesis in synaptic plasticity and diverse functions such as circadian rhythms, social behaviour, memory, and cognitive flexibility[8–13]. While these studies demonstrate the role of initiation factors in PIC recruitment and 5′ leader scanning, they do not resolve the dynamics of SSU scanning and start codon recognition.

Analysing 80S ribosome-bound mRNAs through ribosome profiling and related techniques such as TRAP-seq has provided insight into many questions related to translation control in the mammalian brain[9,14,15]. However, ribosome profiling characterises only the elongation step of translation, such that the dynamics and regulation of translation initiation in the mammalian nervous system have remained elusive. Translation complex profiling (TCP-seq) was the first method to characterise the occupancy

[1]Computational Biology Unit, Department of Informatics, University of Bergen, Bergen, Norway. [2]Michael Sars Centre, University of Bergen, Bergen, Norway. [3]Department of Biomedicine, University of Bergen, Bergen, Norway. [4]Mohn Research Center for the Brain, University of Bergen, Bergen, Norway. [5]Department of Biosciences, University of Oslo, Oslo, Norway. [6]These authors contributed equally: Preeti Madhav Kute, Francois P. Pauzin, Kornel Labun. ✉e-mail: clive.bramham@uib.no; eivind.valen@ibv.uio.no

of SSUs across the transcriptome[16]. TCP-seq, like Ribo-seq, involves digesting the lysates with RNase I and separating 40S and 80S fractions based on the polysome profile obtained after sucrose gradient density centrifugation. RNA from these fractions is then sequenced and analysed. In the original TCP-seq study, only those mRNAs bound by at least an elongating (80S) ribosome were used to explore translation initiation in yeast and a modified technique, ribosome complex profiling (RCP-seq) was developed to capture all mRNAs, including those bound only by a 40S with no elongating (80S) ribosomes, to explore initiation in the early development of zebrafish[17]. In the mammalian brain, however, transcriptome-wide characterisation of translation initiation has not been attempted.

Here, we developed a UV light crosslinking method capable of producing sufficient input material for RCP-seq for brain tissues from mice. This approach generated libraries of similar quality to previous studies[16,17] and allowed a detailed characterisation of translation initiation and elongation in the nervous system for the first time. Using the dentate gyrus region of the hippocampus and the cerebral cortex from mice, we show regulation of translation at the initiation stage. First, SSUs accumulate upstream of the start codon in a poised state on synaptically localised mRNAs, indicating a regulatory step during the transition to the elongation step. Second, our data reveal regulation during scanning with a role for uORF-mediated translational repression of the downstream CDS. Third, we show that neuron-enriched transcripts, relative to their RNA abundance, have a higher recruitment of ribosomes, leading to enrichment of both scanning and elongating ribosomes as compared to glia-enriched transcripts. Finally, we show that neuronal transcripts preferring monosomal translation have reduced recruitment of ribosomes, causing their reduced translation efficiency.

## Results

### Mapping the SSU and 80S ribosomal complexes in the dentate gyrus (DG) and cerebral cortex of the adult mouse brain

Previous methods for capturing scanning SSUs have used formaldehyde and chemical crosslinking for the SSU fixation to obtain footprints on the mRNAs[16–19]. In mouse brain tissue, the use of 0.1% or 0.05% formaldehyde resulted in less crosslinking for polysomes as compared to UV-crosslinked lysates (Supplementary Fig. 1a). Thus, we opted for UV-crosslinking, which has been previously used for studying RNA binding protein (RBP) interactions with mRNAs[20] and for polysome profiling in the brain tissue[21]. The steps to efficiently crosslink the SSU and the 80S on RNAs from the mouse brain tissue are outlined in Fig. 1a and detailed in the methods section. Briefly, the pre-cleared lysate was exposed to UV irradiation and digested with RNase I to obtain footprints (Supplementary Fig. 1b). Since the SSU peak was undetectable in the polysome profiles, RNA was isolated from DG polysome profiling fractions and run on a Bioanalyzer to detect the presence or absence of the 28S rRNA to differentiate SSU from the 80S fractions (Supplementary Fig. 1c, d). The fractions corresponding to the SSU and 80S were collected post-digestion and libraries were prepared and sequenced to 20–130 million reads per library (Supplementary data 2). Total RNA libraries were prepared from matched undigested samples to estimate RNA abundance (Supplementary data 2).

The library quality was comparable to previous studies[16,17] where contaminants were largely from rRNAs (Supplementary Figs. 1e). Of the SSU reads mapped to mRNAs, 52% mapped to the 5′ leader (Fig. 1b) while 94% of the 80S reads mapped to the coding sequence (CDS) (Fig. 1b). Accordingly, when normalised to the expected fraction of reads from a given region, estimated from the total mRNA library, the SSU libraries and 80S libraries were enriched as expected in the 5′ leader and CDS respectively (Fig. 1c). Footprint length distribution for the 80S libraries was enriched within the expected range of 26–30 nt[22] and showed longer reads up to 60 nt, implying either the presence of SSUs with initiation factors as discussed in a previous TCP-seq study[19] or ribosomal interactions with other RNAs such as lncRNAs[23]. Similarly, SSU footprints had a broader distribution with many longer fragments, 20–75 nt in these libraries (Supplementary Fig. 1f) as previously reported[16,17]. Metagene heatmap of SSU footprints over the

translation initiation site (TIS) for the DG showed a range of fragment lengths (20–75 nt) forming a diagonal preceding the TIS (Fig. 1d). This has been attributed to longer pre-initiation complex conformations due to the presence of initiation factors[16,18]. Some SSU footprints are also present internally in the CDS, seen upstream of TTS (Fig. 1e), potentially representing contamination from dissociated 60S subunits during sample preparation or instances of 'leaky scanning' where the PIC scans through the start codon and into the CDS. For the 80S ribosomes (Fig. 1f, g), the footprints are enriched at the TIS and show the expected 3-nucleotide periodicity throughout the CDS. Although the reads mapping specifically to either 5′ leaders or CDS were low for SSU and 80S libraries (Supplementary Fig. 1g) replicates were highly correlated (Supplementary Figs. 1h–j), and clustered separately as SSU, 80S, and RNA (Supplementary Fig. 1k) with PCA, demonstrating that the protocol is robust and reproducible.

To assess the transferability of the technique to another brain region, we applied RCP-seq to cerebral cortex tissue (Supplementary Fig. 2a–d). The library quality was comparable to the DG, with 37% of the SSU reads and 94% of the 80S reads of mRNA reads mapping to the 5′ leaders and CDS respectively (Fig. 2a and supplementary Fig. 2e) with a strong enrichment relative to the expected distribution based on RNA (Fig. 2b). Footprint length distributions were within the expected range of 20–75 nt but more abundant between 25–30 nt for SSU and 28–32 nt for 80S, with some reads going up to 60 nt (Supplementary Fig. 2f). As compared to DG, cortical SSU showed less enrichment of longer footprint lengths (>40 nt), which could be attributed to either a technical difference or biological difference between the two tissue types. As expected, the metagene heatmap of SSU footprints showed a diagonal of fragments (20–75 nt) preceding the TIS (Fig. 2c), which is attributed to a longer pre-initiation complex conformation due to the presence of initiation factors[16,18]. Surprisingly, we observed an additional diagonal pattern from ~10 nt longer reads, indicating a second SSU PIC conformation at the TIS (Fig. 2c). This suggests that additional factors such as eIF3B, eIF4G1 could be involved in the formation of this SSU configuration as discussed previously[18]. Some SSU footprints are also present internally in the CDS (Fig. 2d), as seen in DG (Fig. 1e). For the 80S ribosomes, the footprints are enriched at the TIS and TTS and show the expected 3-nucleotide periodicity throughout the CDS (Fig. 2e, f).

Taken together, our UV-crosslinking method successfully captures the SSU and 80S ribosomal complexes transcriptome-wide in the adult mouse DG region of the hippocampus and in the cerebral cortex.

### Poised SSUs upstream of TIS in selected genes

Further analysis of the start-codon-associated SSU footprints showed two distinct populations of SSUs with different 5′ ends but overlapping 3′ ends (Fig. 3a, Supplementary Fig. 3a, b). The majority of the SSU population had a 5′ end at -12 with 3′ ends at +11 to +25, similar to those observed for SSUs from HEK293T[19]. This SSU position implies a closed conformation of SSU following AUG recognition[16] (Fig. 3b, top row). The longer SSU footprints have previously been explained as pre-initiation complexes associated with initiation factors (IFs) that extend the protection of the RNA[16–19]. These longer SSU footprints with extended 5′ ends at −46 to −36 nt (Supplementary Fig. 3a, b, blue lines) share 3′ ends with the shorter conformation that lack IFs and do not continue scanning further downstream. These are therefore "poised" to initiate elongation (possibly in a post-AUG-recognition conformation). Besides SSUs with IFs (Fig. 3b, middle row) the length of "poised" is also consistent with two queued SSUs (Fig. 3b, bottom row). Previous TCP-seq studies have speculated that these longer SSU fragments can arise due to queuing of multiple SSUs near the start codon[16,19]. Such queued SSUs were also demonstrated to occur in vitro[24] and posited in various studies (summarised in Supplementary Table 1). To rule out that UV-crosslinking specifically enriches these SSU conformations in the brain tissues, we performed formaldehyde crosslinking in HEK293T cells following the previously published TCP-seq protocol[19] and compared it to UV crosslinking. In formaldehyde crosslinked samples, we observed a similar enrichment of SSUs with 5′ ends around -50nt relative to the TIS (Supplementary Fig. 3d), while absence of crosslinking gave no such

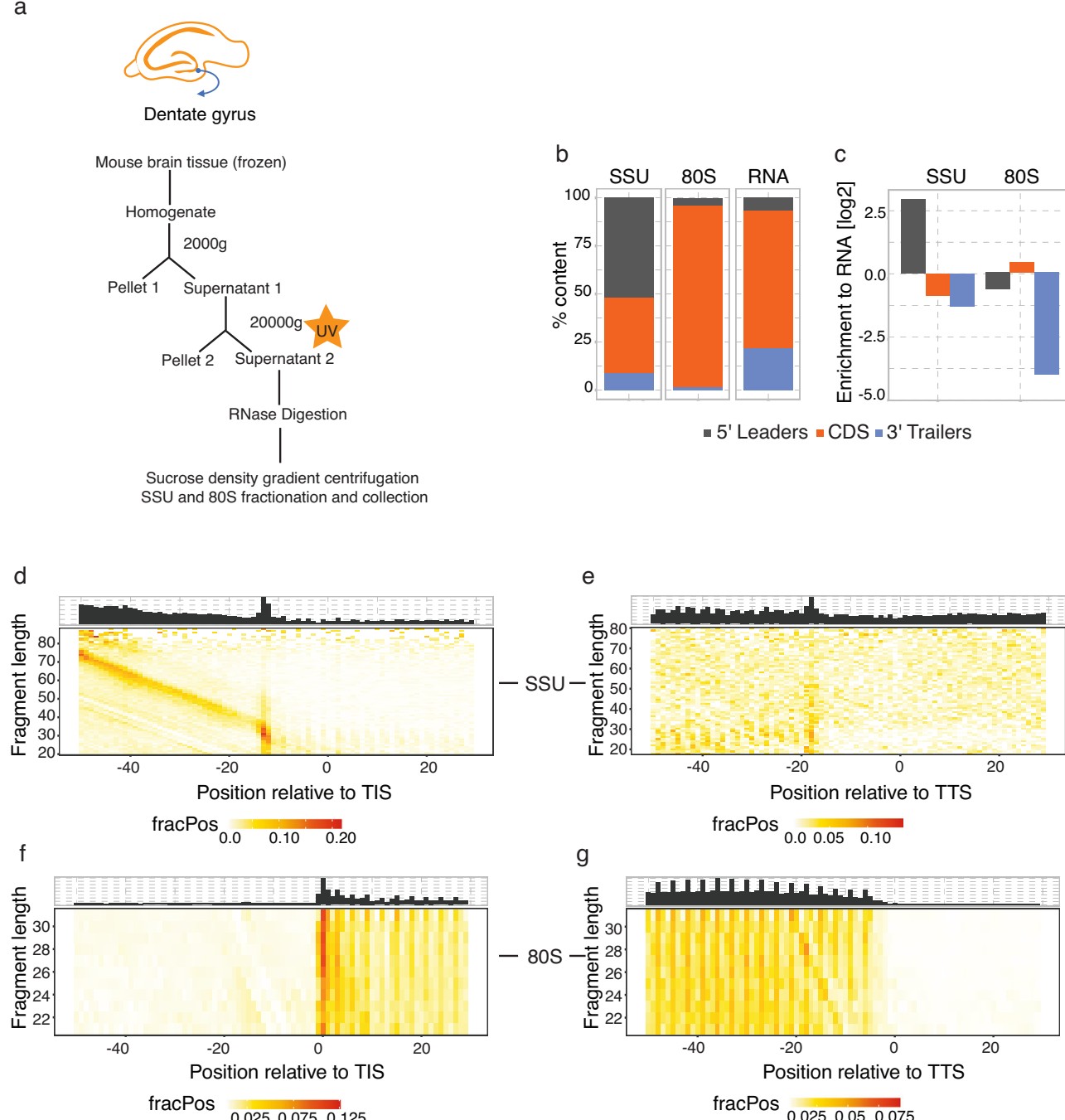

**Fig. 1 | Ribosome complex profiling (RCP-seq) captures small ribosomal subunits (SSU) and elongating ribosomes (80S) from the mouse DG tissue. a** Steps for ribosomal complex profiling using UV crosslinking for the fixation of the SSU and 80S in the DG. **b, c** Content plots displaying the mapping of SSU, 80S, and total mRNA counts to the 5′ leaders, CDS, and 3′ trailers of the transcripts (*n* = 3). **b** shows the % count for each library and **c** displays the normalised counts to the total RNA. Footprint length distribution for the 5′ end of SSU fragments in the DG for **d** translation initiation site (TIS) and **e** translation termination site (TTS). Footprint length distribution for the 5′ end of 80S fragments in the DG for (**f**) translation initiation site (TIS) and (**g**) translation termination site (TTS). FracPos indicates the fraction of counts per position per gene.

enrichment of SSU footprints (Supplementary Fig. 3c). UV crosslinking before and after lysis also showed a similar but lesser enrichment of SSUs with 5′ ends around -50nt and relative to the TIS (Supplementary Fig. 3e–f), indicating that similar SSU conformations are observed both in HEK293T cells and brain tissues, irrespective of the crosslinking method.

We next asked where and to what extent poised SSUs occur in the transcriptome. As poised SSUs overlap the start codon, they cannot coexist on the same transcript with the short SSU conformations at −12 nt that initiate elongation. The transition between these two conformations could therefore indicate a regulatory step before or during start codon recognition concomitant with a change in ribosomal conformational state. We, therefore, defined an "SSU poised ratio" as the ratio between the two mutually exclusive configurations of poised SSUs and initiating SSUs (boundaries in Fig. 3a, b) and calculated this metric for 2725 transcripts that had sufficient coverage of SSU reads, from the cortical and DG tissues. This revealed that most transcripts have more initiating SSUs than poised SSUs, but that a subset of transcripts have substantially higher poising (Fig. 3c below *vs*. above zero). The length of the 5′ leaders can influence the density of SSUs[16]

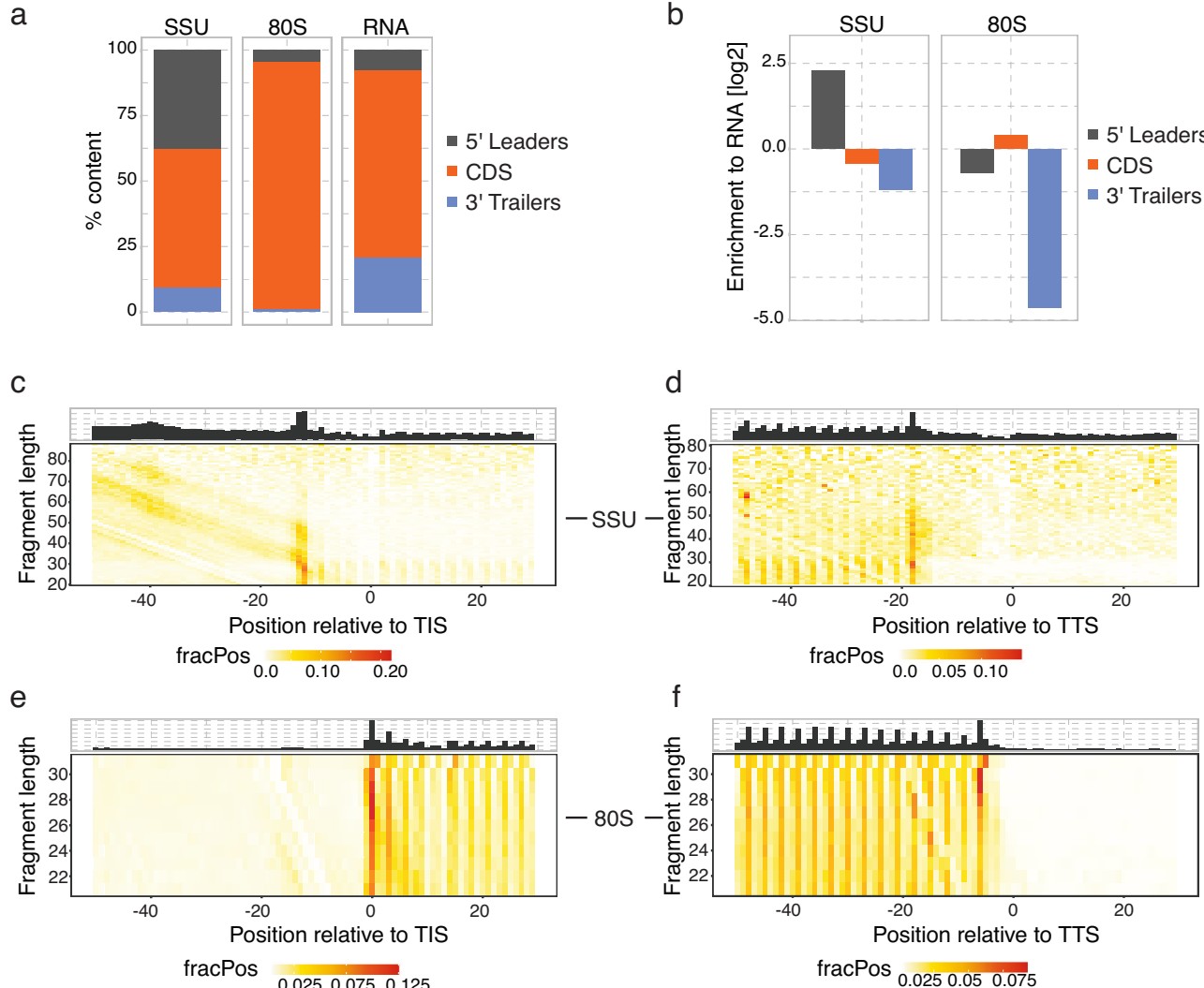

**Fig. 2 | Ribosome complex profiling (RCP-seq) captures small ribosomal subunits (SSU) and elongating ribosomes (80S) from the mouse cortical tissue.** **a**, **b** Content plots displaying the mapping of SSU, 80S, and total mRNA counts to the 5′ leaders, CDS, and 3′ trailers of the transcripts (*n* = 3). **a** shows the % count for each library and **b** displays the normalised counts to the total RNA. Footprint length distribution for the 5′ end of SSU fragments in the cortex for (**c**) translation initiation site (TIS) and (**d**) translation termination site (TTS). Footprint length distribution for the 5′ end of 80S fragments in the cortex for (**e**) translation initiation site (TIS) and (**f**) translation termination site (TTS). FracPos indicates the fraction of counts per position per gene.

that could cause SSU accumulation on shorter 5′ leaders. Interestingly, the length of 5′ leaders was not a contributing factor to the extent of SSU poising (Supplementary Fig. 4a). We selected the strongest SSU-poised transcripts as those having a two-fold enrichment in poised *vs.* initiating SSUs (76 genes for DG and 198 for cerebral cortex) and the weakest as two-fold depletion (1536 genes for DG and 1041 for cerebral cortex). This revealed that poised genes have substantially more scanning activity with a higher occupancy of SSUs across the 5′ leader, but a weaker presence of 80S ribosomes at the TIS (Fig. 3d, e). This could indicate increased recruitment of PIC to the RNA, potentially combined with either a slower joining of 60S or a faster translocation of 80S away from the TIS. We calculated the translation efficiency (TE, see Methods) metric to assess the presence of 80S on these transcripts and observed higher TE for transcripts with a high poising ratio as compared to those with a low poising ratio for DG but not for the cerebral cortex (Fig. 3f, g).

To probe the biological relevance of poised SSUs, we analysed the 42 genes poised in both DG (76 genes) and cortex (198 genes) tissues for overrepresented functional GO categories against a restrictive background of other genes with sufficient reads, in total 2725 genes (1917 DG, 1969 cortex) (Supplementary data 3). This revealed four enriched categories

(Fig. 3h), all belonging to the synaptic compartment of the neuron. A prominent gene is *Camk2α* (Fig. 3i, j), where the SSU occupancy plot showed a strong enrichment of SSUs near and up to 60 nt upstream of the TIS (Fig. 3j). Other examples include *Clstn3*, *Pcbp2*, *Pacsin1*, *Rtn3*, *Prnp*, and *Prkcg* (Supplementary Fig. 3g–l). Together this shows that poised SSUs are enriched in a subset of functionally related genes.

### uORF-mediated regulation in the DG tissue

Upstream open reading frames (uORFs) are regions in the 5′ leaders of mRNAs with a start codon and a downstream in-frame stop codon, which could potentially encode a peptide. While many of these uORFs are translated, it is believed that most of these do not produce functional peptides but rather contribute to regulating translation of the downstream CDS[25,26]. A key regulatory feature of uORFs is that some ribosomes detach after translating the uORF and are therefore unable to initiate at the protein-coding TIS (Fig. 4a), leading to translation downregulation of the CDS. To explore the role of uORFs in the translation regulation of downstream CDSs in the DG, we defined uORFs as ORFs with 80S read coverage and having an AUG as the start codon. For those 5′ leaders containing multiple eligible uORFs we selected the one with the highest read coverage (totalling 1062, see methods).

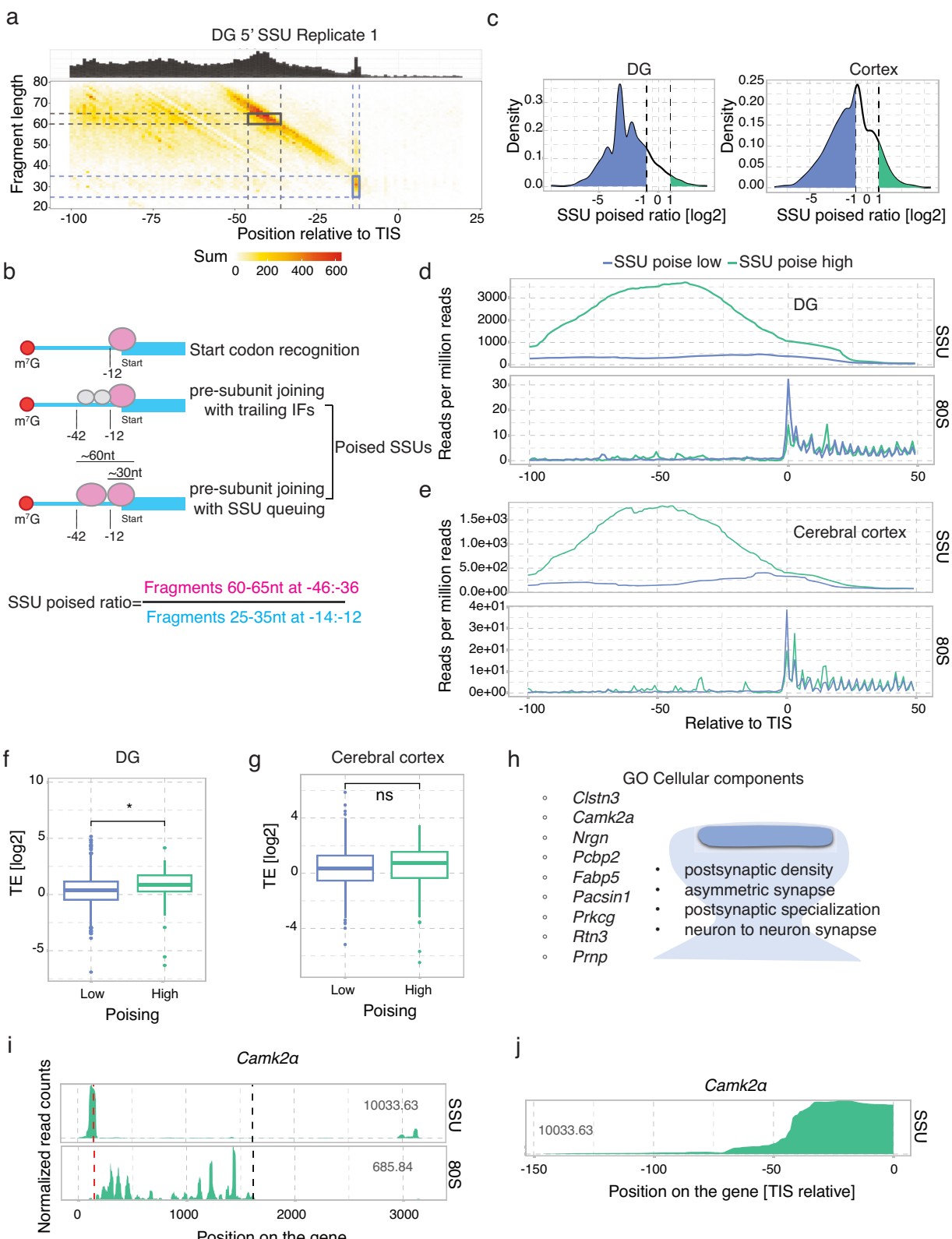

These were termed "active uORFs" (Supplementary data 4). To check the effect of the translation of active uORFs on downstream CDS, we looked at the effect of the presence or absence of uORFs on the translation of the transcripts. This showed a modest but significant inhibitory effect of having actively translated uORFs (Fig. 4b). To further explore the role of uORFs in regulation of CDS translation we calculated the scanning efficiency (SE)

(Fig. 4c) which quantifies the abundance of SSU occupancy in the leaders normalised to the RNA levels. SEs were comparable between transcripts harbouring uORFs and those without, implying that uORFs do not globally have a significant effect on the overall occupancy of SSUs over the leaders (Fig. 4d). On the other hand, uORFs can affect the SSU traversal over the 5' leaders either through slowing them down by the relatively slower process of

**Fig. 3 | SSU build-up observed in the leaders of transcripts in the DG and the cerebral cortex. a** Footprint length distribution for the 5′ end of SSU fragments in the DG for the TIS, highlighting initiating SSUs (at −12 nt) and poised SSUs (−46: −42 nt). **b** Schematic of different conformations of SSUs near the TIS of transcripts and formulae for calculating SSU poised ratio. **c** Distribution plot for SSU poised ratio (log$_2$) for DG and cortical transcripts. Dotted lines indicate SSU poised ratio cut-off of −1 and +1. Coloured regions indicate transcripts with lower (blue) or higher (green) poised ratios from the cut-off lines. **d** Metagene coverage for transcripts with SSU poised ratio below (blue) or higher (green) than the cut-offs in the DG relative to the TIS. **e** Metagene coverage for transcripts with SSU poised ratio below (blue) or higher (green) than the cut-offs in the cerebral cortex relative to the TIS. **f** Log$_2$ of Translation Efficiency (TE) for genes with high or low poised ratios (FPKM-based, t-test statistics) for DG transcripts. **g** Log$_2$ of Translation Efficiency (TE) for genes with high or low poised ratios (FPKM-based, t-test statistics) for cortical transcripts. **h** GO enrichment categories for genes with SSU poising ratio more than 2 and the list of genes enriched in the categories in both DG and cortex. **i** Single gene profile for SSU and 80S, without intronic regions, and plotted for the longest isoform of *Camk2α* (Calcium/calmodulin-dependent protein kinase type II subunit alpha) from DG. Numbers indicate FPKM values. Dotted lines indicate TIS (red) and TTS (black). **j** Single gene profile for SSU within the 5′ leader region of *Camk2α* from DG. Numbers indicate FPKM values.

elongating a uORF or by dissociating a proportion of them from the RNA after uORF translation. To check whether uORFs had any impact on the number of SSUs reaching the TIS, we looked at poised SSUs at the position -40 nt and length 60 nt for genes with and without uORFs. The poised SSUs reads were normalised to RNA to accommodate any differences at the RNA level. This analysis revealed that the poised SSUs in those genes containing an active uORF were significantly lower (Figs. 4e, f) suggesting that uORFs may act to slow down or disassociate SSUs on their way to the CDS. We next asked whether the presence of uORFs leads to SSU poising at the TIS of the uORF. However, we did not observe this to the same extent as for protein-coding TISs (Supplementary Fig. 4b). This may be because uORF initiation is typically more stochastic than CDS initiation and that leaky scanning (i.e. not initiating at the uORF TIS) may lead to SSUs spread between, potentially, multiple uORFs and the CDS.

### The translational landscape of neuronal and glial genes

The mammalian brain consists of both neuronal and non-neuronal cells, such as astrocytes, oligodendrocytes, microglia, and blood vessel cells. To elucidate the translational landscape of cell-type-specific transcripts, we filtered our dataset according to previously defined neuronal and glial-enriched transcripts[27]. Neuronally-enriched transcripts show more coverage of SSU, 80S and RNA as compared to glial-enriched transcripts, which may be reflective of the cell-type abundances in the tissue (Fig. 5a). Taking into account these differences by normalising by RNA abundance, neuronal-enriched transcripts showed modest but significantly higher TE and SE values as compared to glial-enriched transcripts (Fig. 5b, c). The gene profiles for neuronal transcripts *C1ql2* and *Prox1* (Fig. 5d, e) exemplify the differences in SSU and 80S occupancy compared to glial-enriched transcripts *Tgfb2* and *Fth1* (Fig. 5f, g). Taken together, the neuronal-enriched transcripts are scanned and translated more efficiently than glial-enriched transcripts. This suggests a cell-type-specific difference, with enhanced translation of neuronal transcripts relative to glial transcripts in the DG.

### Scanning and elongation in neuropil-enriched monosome- versus polysome-preferring transcripts

Since translation in neurons is compartmentalised, occurring even in distal dendrites, we next studied transcripts that are differentially localised and translated between the soma and the synaptic neuropil. In a previous study based on ribosome profiling, more than 800 transcripts were found to be predominantly translated in the neuropil (neuronal dendrites and axons) of microdissected rat hippocampal tissue slices[27]. Surprisingly, a recent report based on polysomal profiling of microdissected hippocampal CA1 tissue showed that most of the neuropil translation is mediated by monosomes, with only one actively translating ribosome per transcript[28] (Fig. 6a). The authors also noted that monosome-preferring transcripts had lower translation initiation and elongation rates. Here, we used RCP-seq to directly assess initiation and elongation in neuronal-specific transcripts preferring monosomal or polysomal translation[27,28]. Notably, for both soma and neuropil compartments, monosome-preferring transcripts showed lower SSU, 80S, and RNA abundance as compared to those that preferred polysomal translation (Fig. 6b, c). Interestingly, while monosome-preferring transcripts showed lower TE than polysome-preferring transcripts from the neuropil compartment (Fig. 6d), they displayed similar levels of occupancy

from scanning ribosomes (Fig. 6e) coupled with a lower level of initiation (Fig. 6f).

### Discussion

This study presents the first map of translation initiation dynamics in tandem with elongation in the mammalian brain. This was accomplished by adapting and optimising several steps of our previously published RCP-seq protocol for developing zebrafish embryos[17]. While previous studies mapping SSUs to the mRNAs have used PFA and chemicals such as dithiobis succinimidyl propionate [17,29], the use of low (0.05%) PFA concentrations in the brain tissues gave poor yield of polysomes, possibly due to insufficient lysis of the tissue. To increase the amount of SSUs and to avoid the use of glycine to neutralise formaldehyde, we crosslinked ribosomal complex proteins to RNA using UV irradiation, as used previously to map RBP binding sites and for polysome profiling[20,21]. A possible drawback of UV crosslinking of lysates could be the loss of SSUs during the lysis step. However, in our data, we do not observe any indication of loss and see a distribution of SSUs over the leaders similar to previous studies[16,17,29]. UV irradiation is also known to cause ribosome stress response[30,31], but this study exposes the lysates to UV and thereby bypasses the harmful effects of UV irradiation on tissue. Together these steps enabled us to successfully capture the transcriptome-wide occupancy of both scanning and elongating ribosome complexes from the dentate gyrus and the cerebral cortex region of the mouse brain.

Our libraries of ribosomal occupancy from brain tissues were consistent with previously published datasets in terms of transcriptomic mapping, footprint length distributions, and 80S periodicity[16,17,18]. Accordingly, we observed the diagonal preceding the TIS in the SSU libraries, which has been observed in RCP-seq and TCP-seq studies[17,29]. Interestingly, we observed a second diagonal in the cerebral cortex, indicating an approximately 10-nucleotide-long conformation present at the TIS. While sequencing data alone is insufficient to determine the factors present in this conformation, the read positions and lengths are indicative of the presence of one or more additional initiation factors downstream of the pre-initiation complex, such as helicase eIF4A and its cofactor eIF4B, as shown by recent structural evidence[32].

A major finding of this study was the characterisation of a distinct poised configuration of SSUs enriched at the TISs of functionally related transcripts. Previous studies from yeast, developing zebrafish embryos and HEK293T cells have shown similar SSU patterns[16,17,19], but not in HeLa cell line[18]. This could imply that poising is a species or context-specific phenomenon. We introduced a metric to quantify this configuration relative to the initiating SSUs and showed that SSU poising is highly enriched in mRNAs found in the synaptic compartment, suggesting regulation of the transition from poised to initiating SSUs in a subset of transcripts. In the mouse brain, SSU poising could be necessary for transcripts with slow elongation or start codon recognition rates and could allow fast and abundant translation in response to stimuli, such as during neuronal activity-induced bursts of protein synthesis. Supporting the latter hypothesis, we find that genes with high SSU poising ratios show higher translation efficiency as compared to those with lower SSU poising. One of the interesting gene candidates is *Camk2α*, which is known to undergo rapid dendritic translation upon synaptic stimulation and regulate excitatory synapse

**Fig. 4 | uORF expression in the DG. a** Illustration explaining the role of uORFs on the translation of downstream CDS. **b** Comparison of CDS translation efficiency (TE) in relation to the presence or absence of active uORFs in DG (FPKM-based, *t*-test statistics). **c** Illustration explaining the role of uORFs on the scanning efficiency (SE) of downstream CDS. **d** Comparison of the SE in relation to the presence or absence of active uORFs in DG (FPKM-based, *t*-test statistics). **e** Illustration explaining the role of uORFs on the SSU poising over 5′ leaders. **f** Comparison of the poised SSUs in relation to the presence or absence of active uORFs in DG (FPKM-based, t-test statistics).

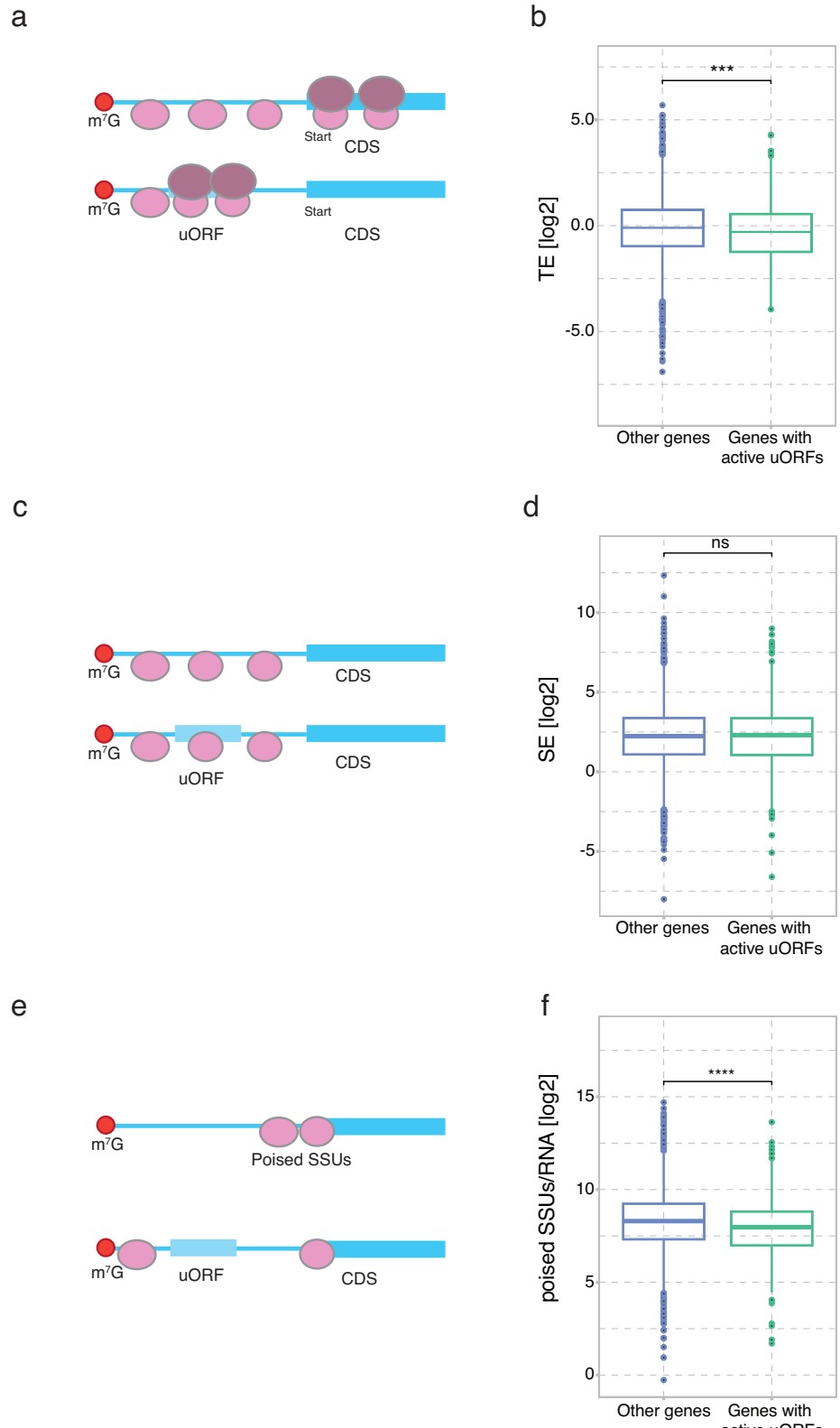

structure and function[33–37]. Thus, the SSU poised ratio could potentially reveal transcripts that are regulated at the stage of start codon recognition.

As discussed in an earlier study[16], factors influencing and causing poising of SSUs may be multifaceted (e.g., mRNA sequence and structure, presence of IRES, SSU pausing). We found that the presence of uORFs is associated with less SSU poising, potentially by disassociating the SSUs after translation of the uORF. Similarly, the presence of uORFs can have an

inhibitory effect on the translation of the downstream CDS. In line with previous reports[38–40], we observed globally a significant decrease in translational efficiency for transcripts harbouring a translated 'active' uORF.

When characterising previously published lists of neuronal and glial enriched transcripts[27], we observed that neuronal transcripts have higher coverage of SSU and 80S than glial transcripts, potentially indicating abundance of neuronal over glial cells in the DG tissue, as also shown

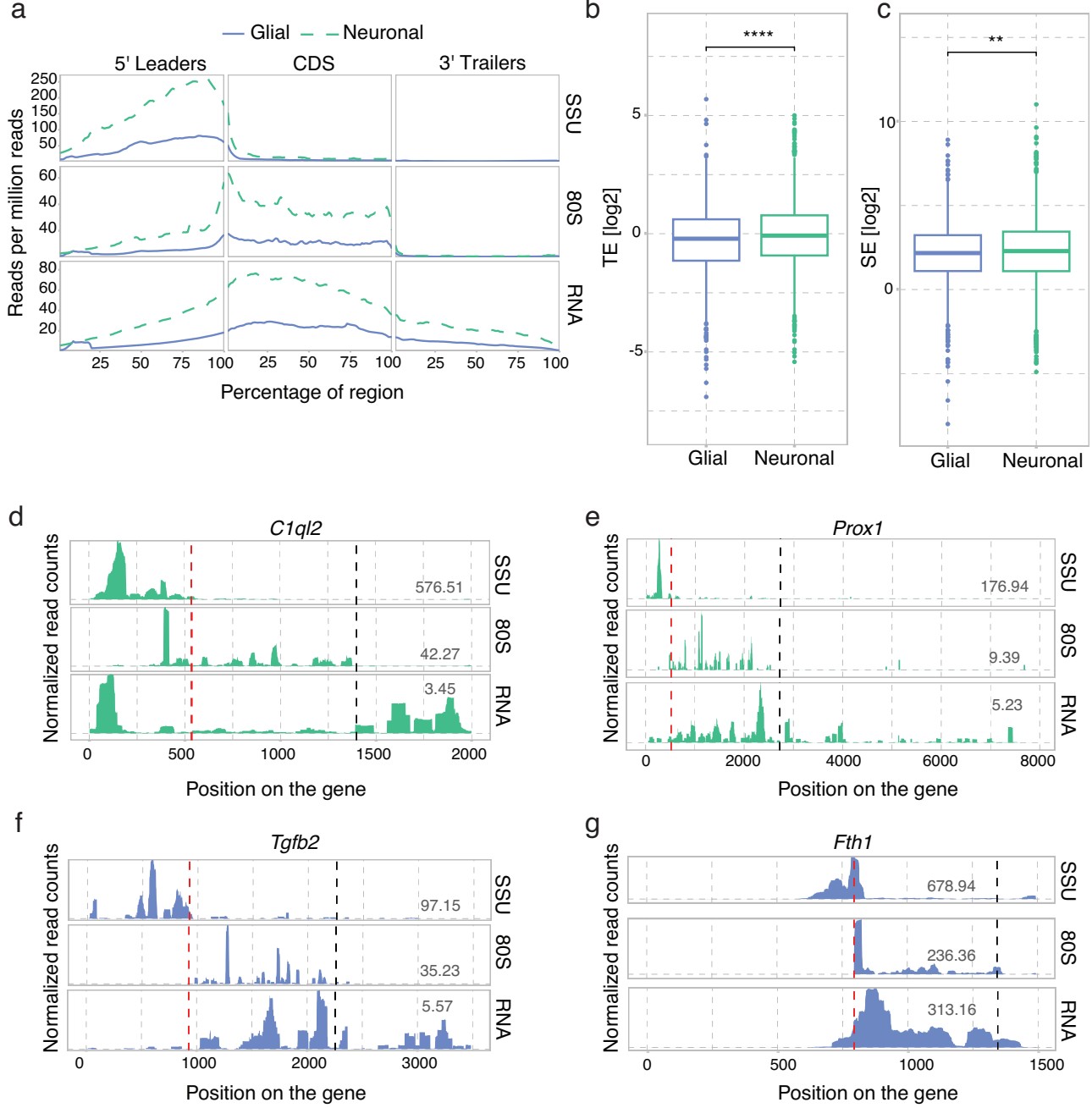

**Fig. 5 | Transcriptional and translational profiles for neuronal and glial-specific genes in the DG. a** Metagene coverage for the SSU, the 80S and total RNA between neuronal- and glial-enriched transcripts. Translation efficiency and scanning efficiency for (**b**) glial and (**c**) neuronal-specific transcripts ((FPKM based, *t*-test statistics). Single gene profiles for SSU, 80S and total RNA coverage, without intronic regions, and plotted for the longest isoform. **d** *C1ql2* (Complement Component 1, Q Subcomponent-Like 2), **e** *Prox1* (Prospero homeobox protein 1), **f** *Tgfb2* (Transforming growth factor beta 2), **g** *Fth1* (Ferritin heavy chain 1). Numbers indicate FPKM values. Dotted lines indicate TIS (red) and TTS (black).

before[41]. However, relative to RNA, neuronal genes showed higher translation and scanning efficiencies than those belonging to the glial category, metrics that are independent of cell-type abundance. Focusing on transcripts that are predominantly monosomal or polysomal provided further mechanistic details on their translation regulation. Neuronal transcripts preferring polysome translation showed higher SSU and 80S occupancy, whereas monosome-preferring transcripts showed lower SSU and 80S reads, implying reduced recruitment of PIC to these transcripts in order to maintain a low initiation and elongation rate. Indeed, monosome-preferring transcripts showed lower initiation rates along with lower translation efficiencies, as compared to polysome-preferring transcripts, suggesting that monosome-preferring transcripts are more sparsely or selectively translated

through a mechanism reducing PIC recruitment. The findings underscore the importance of transcriptome-wide mapping of both initiating and elongating ribosomes in providing an understanding of the plausible translation control mechanism of a specific pool of mRNAs.

## Conclusion

Here, we adapted and optimised RCP-seq for brain tissue, allowing the capture of SSUs, and analysis of translation initiation dynamics in mouse dentate gyrus and cerebral cortex. In tandem with total RNA and 80S analysis, we uncover cell type and transcript-specific regulation at the scanning and elongation stages. We discover thousands of active uORFs associated with repressive function in the translation of the CDS.

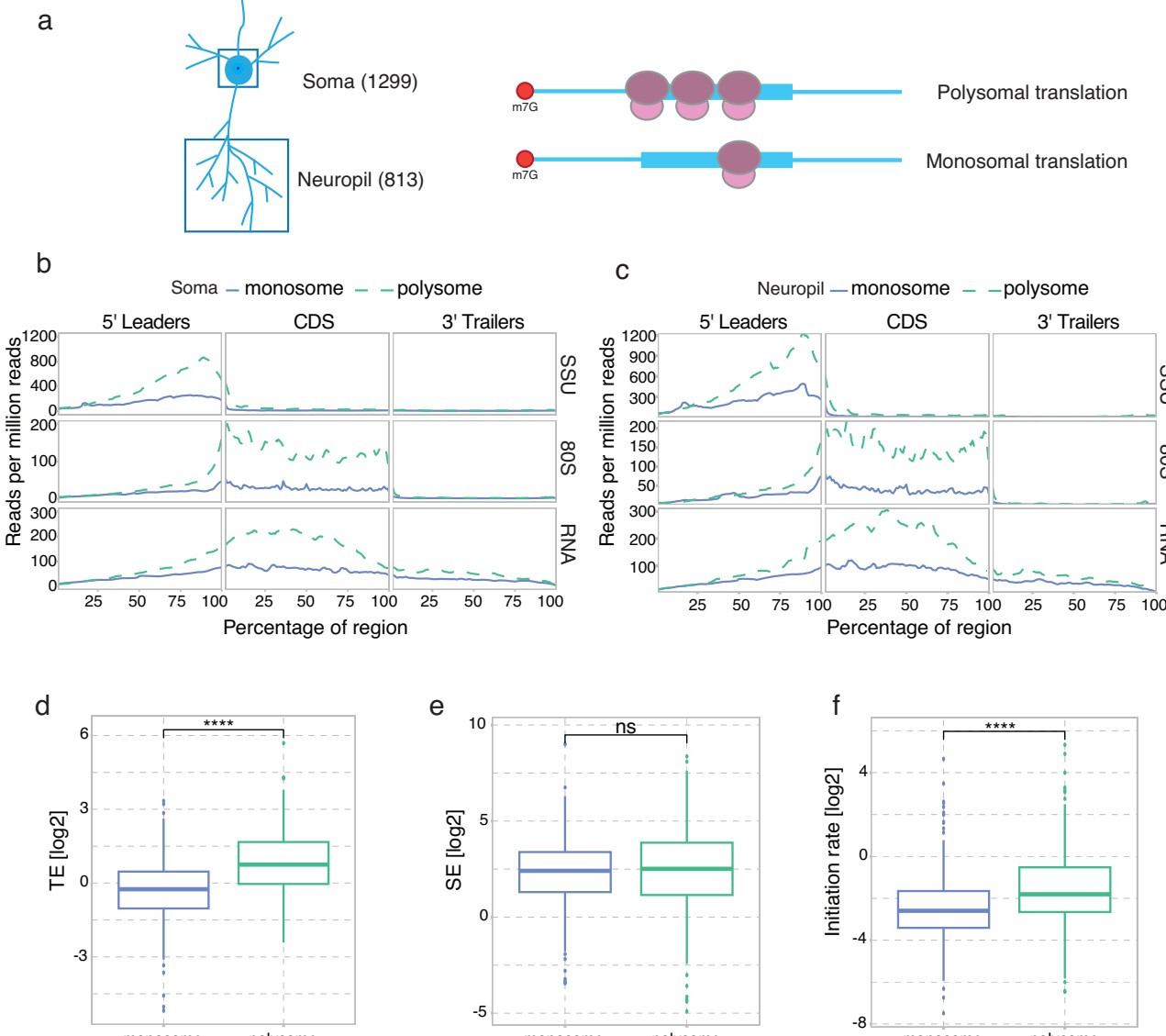

**Fig. 6 | Translation landscape of neuropil and soma-enriched transcripts.**
**a** Representation of neuronal compartments (soma and neuropil) with polysomal translation (multiple ribosomes elongating simultaneously) and of monosomal translation (only one ribosome elongating on the transcript). Numbers in parentheses indicate the number of genes used for analysis. **b** Metagene coverage for transcripts enriched in the soma and preferring monosomal or polysomal translation. **c** Metagene coverage for transcripts enriched in the neuropil and preferring monosomal or polysomal translation. **d** Log$_2$ TE (FPKM based, *t*-test statistics) for neuropil-enriched transcripts between the transcripts preferring monosomal or polysomal translation in the DG. **e** Log$_2$ SE (FPKM based, *t*-test statistics) for neuropil-enriched transcripts between the transcripts preferring monosomal or polysomal translation in the DG. **f** Log$_2$ Initiation Rate (FPKM-based, *t*-test statistics) for neuropil-enriched transcripts between the transcripts preferring monosomal or polysomal translation in the DG.

Additionally, we characterise a phenomenon of poised SSUs upstream of the TIS of synaptically enriched RNAs, identifying it as a possible regulatory step during translation, underlying synapse maintenance and plasticity.

We anticipate that studies on scanning subunit occupancy will help to resolve long-standing questions on the role of initiation factors in the scanning process and how these influence the regulation of individual genes in mammalian brain function and dysfunction.

## Materials and methods
### Animals
12 to 14-week-old C57BL/6 wild-type male mice were used. Mice were bred and housed in their home cages. Room temperature (22( ± 1 °C)) and relative humidity (46 ± 5%) were maintained. Mice had free access to water and food and were maintained on a 12 h light/dark cycle. This research is approved by the Norwegian National Research Ethics Committee in compliance with EU Directive 2010/63/EU, ARRIVE guidelines. We have complied with all relevant ethical regulations for animal use.

### Crosslinking of HEK293T cells and ribosome complex profiling
HEK293T cells were kindly provided by the lab of Dr. Nils Halberg at the Department of Biomedicine, University of Bergen. Cells were cultured and maintained in DMEM supplemented with 10% FBS, penicillin and streptomycin and L-glutamine. Cells are grown in Ø 15 cm dishes up to 60–70% confluency prior to fixation. ~3 h before lysis the volume of media is made up to 20 mL.

For in vitro UV crosslinking, cells are UV cross-linked at 254 nm with 400 mJ/cm$^2$ three times in a CL-1000 ultraviolet crosslinker (UVP). The plate is swirled between each round of irradiation for uniform crosslinking. Cells are then washed with 25 mL of ice-cold 1×PBS. The cells are scraped in 1000 μL of ice-cold lysis buffer (10 mM HEPES-KOH pH 7.5, 62.5 mM KCl, 2.5 mM MgCl$_2$, 1 mM DTT, 1% Triton ×100, and protease inhibitor

cocktail). The lysate is incubated in ice for 10 min and spun at 14,000 RPM for 5 min to obtain the supernatant.

For formaldehyde crosslinking, 600 μL of 10% formaldehyde (0.3% final concentration) is added by tilting the dish to the pool of media and turning the dishes back to an even level. The cells are incubated on ice for 5 min with intermittent swirling. Formaldehyde is quenched by adding 600 μL of chilled 2.5 M glycine to media and incubated for 5 min, gently swirling after every minute. Cells are then washed with 25 mL of ice-cold 1X PBS. The cells are scraped in 1000 μL of ice-cold lysis buffer. The lysate is incubated on ice for 10 min and spun at 14,000 RPM for 5 min to obtain the supernatant.

For UV crosslinking after lysis, cells were washed with 20 mL ice-cold 1×PBS and scraped in a 1000 μL lysis buffer. The lysate was incubated in ice for 10 min and spun at 14,000 RPM for 5 min. The supernatant in a 60 mm dish is then UV cross-linked at 254 nm with 400 mJ/cm$^2$ three times in a CL-1000 ultraviolet crosslinker (UVP). The dish was swirled between each round of irradiation for uniform crosslinking.

For RCP-sequencing, 10 OD lysates are digested with 70U of RNase1 for 45 min at 24 °C. The digestion is stopped with 28U of Superase In. The lysates are layered on 15–45% sucrose gradients and spun at 39,000 RPM for 4 h at 4 °C. Twenty fractions are collected, of which RNA is extracted from fractions corresponding to 40S peak. De-crosslinking buffer (1% SDS, 10 mM EDTA, 10 mM Tris-HCl (pH 7.4), 10 mM glycine) is added to the 40S fractions. An equal volume of Phenol-Chloroform (pH 4.5) is added to the above mix, and samples are incubated at 65 °C for 45 min at 1300 RPM shaking. The final RNA pellet is air-dried and resuspended in 12 μL of nuclease-free water. RNA is processed according to the protocol mentioned later in the section.

### UV fixation of brain tissue and polysome profiling
Animals were anaesthetised with urethane (1.5 g/kg) and then sacrificed immediately. The whole brain was removed and positioned on a filter paper placed on a glass plate cooled with ice. After isolating both hippocampi, blood vessels and connective tissue were removed. The Cornu Ammonis region was separated from the dentate gyrus (DG) before placing the DG in microtubes in dry ice. After isolating both cerebral cortices, the white matter was carefully removed before placing the entire cortices in microtubes in dry ice. The brain samples were given a liquid nitrogen bath and then placed at −80 °C for storage. It took a maximum of 10 min from sacrifice to sample storage to preserve RNA integrity. Long-term storage of tissue at −80° C was avoided to prevent RNA degradation.

Brain tissue was washed twice with ice-cold 1XPBS containing cyclo-heximide (0.1 mg/mL) before proceeding with the fixation/homogenisation. For formaldehyde fixation, whole tissues were incubated with 1 mL for-maldehyde solution for 5 min on ice. To neutralise formaldehyde, 1 M glycine was added and incubated for another 5 min. The tissue was then washed with 1×PBS twice before homogenisation. For UV crosslinking, the tissue was first lysed, and the lysate was exposed to UV. Tissue homo-genisation was done in lysis buffer (50 mM Tris-Cl, pH 7.4; 100 mM KCl; 5 mM MgCl$_2$; EDTA-free protease inhibitor cocktail; 1 mM DTT; 0.1 mg/mL cycloheximide, 40 U/mL Superase Inhibitor) in a dounce homogeniser with 15 strokes of loose piston and 15 strokes of the tight piston on ice. The volume of the lysis buffer used depended on the size of the tissue. For the cortex (one hemisphere), 0.5 mL was used. For DG regions, 0.3 mL was used. For each biological replicate, 1 cortical hemisphere (left/right) (~17 μg RNA) was used. Thus, three biological replicate experiments were per-formed from cortical hemispheres (left/right) of three different mice. For DG tissue, 10 DGs (left + right) (~9 μg RNA) from 5 mice were pooled for one biological replicate. Thus, three biological replicates were performed from 15 different mice. Both the cortical and DG tissues were isolated from the same mice. The homogenate was transferred to an RNase-free 1.5 mL tube and 1% NP40 was added to the homogenate, mixed well, and incubated in ice for 10 min. The homogenate was spun at 2000$g$ for 10 min at 4 °C. The clear supernatant (S1) was collected and then spun at 20,000 $g$ for 10 min at 4 °C. The clear supernatant (S2) obtained was transferred to a 3.5 cm tissue

culture dish placed on a bed of ice slush. The supernatant was UV cross-linked at 254 nm with 400 mJ/cm$^2$ three times in a CL-1000 ultraviolet crosslinker (UVP). The plate was swirled between each round of irradiation for uniform crosslinking. The RNA concentration was estimated by the absorbance using both qubit and nanodrop. This can be done before or after UV irradiation. The lysate was layered over a 10–50% sucrose gradient (10 mM Tris-Cl pH7.4, 100 mM KCl, 5 mM MgCl$_2$, 2 mM DTT) and spun at 40000 RPM for 2–4 h at 4° C in an ultracentrifuge. The gradients were then run through a UV detector-fractionator (Biocomp).

### Ribosome complex profiling: library preparation and sequencing
RNase digestion was done as previously described[42]. 5U of RNase 1 was used for one unit of absorbance at A260 for a total volume of 300 μL brain lysate. RNase digestion for UV crosslinked lysates was done at 25° C for 45 min. The digestion was stopped by the addition of 50U of Superase Inhibitor. For total RNA sequencing, 50–100 μL of lysate was kept aside before the digestion step. 10–50% sucrose gradients were made 45 min before the ultracentrifugation in polycarbonate tubes (Science Services, Germany, S7030). 450–500 μL of digested lysates were then layered over the gradients and run in SW41 Ti rotor at 40,000 rpm for 3 h. The gradients were then run through a UV detector-fractionator (Biocomp) and 20 fractions were col-lected. Fractions corresponding to the 40S and 80S peaks were used for RNA extraction and library preparation.

RNA extraction was done using TRIzol LS, and in the final step, the RNA pellet was resuspended in 10 μl of nuclease-free water. The RNA sample was then run on a 15% polyacrylamide gel for size selection for the 40S and 80S footprints using RNA denaturing dye, with an ultra-low DNA ladder as the marker (Invitrogen: 10597012). For 40S footprints, gel from 20–80 nt was cut and for 80S footprints, gel from 20–50 nt was cut. The gel pieces were crushed and resuspended in 500 μL of 0.3 M Sodium acetate solution overnight, at 4° C, and then filtered through 0.4 μm cellulose filters. RNA was precipitated overnight at −20° C in the presence of 2 μl of gly-coblue (Life Technologies: AM9516) and 375 μL isopropanol and resus-pended in 10 μl of nuclease-free water. The size-selected footprints were 3′ dephosphorylated using PNK for 2 h at 37° C in a final volume of 20 μl. After dephosphorylation, the RNA was concentrated and cleaned by the Oligo clean and concentrator kit (Zymo Research) in a final volume of 14 μL in nuclease-free water. The RNA footprints were then rRNA depleted with oligos from siTOOLS (riboPOOL riboseq h/m/r) and eluted in a final volume of 10 μl. Small RNA libraries were made using the Takara SMARTer smRNA kit and the manufacturer's protocol was followed. PCR cycles were based on the starting concentration of RNA. Total RNA libraries were prepared using the Takara SMARTer total RNA HI mammalian kit and the manufacturer's protocol was followed. Libraries containing primers or primer dimers were cleaned up using RNA XP clean-up beads (catalogue #A63987). The size and molarity of the libraries were estimated by obtaining bioanalyzer profiles (Agilent). Libraries were run as 100 bp single-end on the NOVAseq 6000 platform up to a depth of 100 M reads for each small RNA library or 30 M reads for the total RNA library.

### Ribosome complex profiling: analysis
The repository at https://git.app.uib.no/valenlab/brain_dg_cortex_rcp and https://git.app.uib.no/valenlab/preeti_40s_2025 contains all the code that was used for analysis and all of the processed data and figures are available there for inspection. All figures and processed tables are present in the repository. For the data processing, we used ORFik (1.19.3) as shown in the script '0_preprocess.R'. Analysis uses the latest at the time, Gencode mouse release M31 (GRCm39 v107). ORFik pipeline uses fastp software for trimming and STAR for alignment. Paired-end fastq reads are initially aligned to the contaminants-phix, rRNA, ncRNA, tRNA, and finally to the genome. Important options that were set are adaptor.sequence = 'AAAAAAAAAA', trim.front = 3, min.length = 20. Aligned data are pro-cessed using ORFik and custom scripts available in the data repository. We restricted genes with multiple transcripts to a single transcript by selecting the one with the longest coding sequence. For statistical testing, we used

cutoffs of log2 fold change of 1 or −1 and 0.05 as the significance level. Upstream ORFs were detected using the ORFik function 'findUORFs' which by default searches for HTG start codons (not GAG), but we used only ATG as start codons. Furthermore, we filtered out these uORFs that don't have any values over SSU/80S/RNA. We restricted further to one uORF per transcript by selecting the one with the highest translational efficiency.

For comparisons at the global level, we calculate measures produced per library using FPKM values instead of raw counts and average replicates, furthermore, we normalise using reads per million reads for that particular group of genes to gain a comprehensive perspective. As we can see below, uORF measures are adapted from gene measures by changing the perspective of CDS to that of the uORF, and 5′ leader and 3′ trailer are becoming regions upstream/downstream of uORF CDS, but within boundaries of the transcript mRNA.

## Formulae used in this study

*Scanning efficiency (SE) = SSU on leaders/RNA on mRNA (leader + CDS + trailer)*

*Translation efficiency (TE) = 80S on CDS/RNA on mRNA*

*Initiation rate (IR) = 80S on CDS/SSU on leaders*

*uORF TE = 80S on uORF/RNA on uORFs*

*uORF SSU consumption rate = SSU upstream of uORF/SSU downstream of uORF*

*uORF ratio = 80S of uORF/80S of CDS*

*SSU poising: Long SSU (60–65nt) at −46: −36 relative to TIS/Short SSU (25–35nt) at −14: −12 relative to TIS*

*Poised SSUs/RNA: Long SSU (60-65nt) at −46: −36 relative to TIS/RNA on mRNA*

## Statistics and reproducibility

Animal experiments were done in three independent biological replicates for each tissue ($n = 3$). For each biological replicate, 1 cortical hemisphere (left/right) (~17 µg RNA) was used. Thus, three biological replicate experiments were performed from cortical hemispheres (left/right) of three different mice. For DG tissue, 10 DGs (left + right) (~9 µg RNA) from 5 mice were pooled for one biological replicate. Thus, three biological replicates were performed from 15 different mice. Both the cortical and DG tissues were isolated from the same mice. No group comparison was done in this study. No experimental units or datapoints were excluded in this study. Only one replicate was performed for HEK293T experiments ($n = 1$). For boxplot figures, *t*-test statistics were used to compare means of the distributions with corrected *p*-values using Benjamini–Hochberg method. Statistical analysis was done using R and ggpubr package.

## Reporting summary

Further information on research design is available in the Nature Portfolio Reporting Summary linked to this article.

## Data availability

Supplementary Data 1–4 are available as supplementary data files. Sequencing raw reads are published under accession number PRJEB72224 in the European Nucleotide Archive.

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

## Acknowledgements
F.P.P. was funded by the L.Meltzers Høyskolefond 2022— Ekstratildeling— for the project "Molecular control of protein synthesis underlying the plasticity of neural communication in the mammalian brain' (Project #103517122). Research in the CRB lab is supported by the Trond Mohn Research Foundation (TMS2021TMT04). P.K. and E.V. are funded by the Norwegian Cancer Society (Project #190290). The Genomics Core Facility (GCF) at the University of Bergen, which is a part of the NorSeq consortium, provided services on sequencing of small and total RNA libraries on NOVAseq 6000 platform; G.C.F. is supported in part by major grants from the Research Council of Norway (grant no. 245979/F50) and Bergen Research Foundation (BFS) (grant nos. BFS2017TMT04 and BFS2017TMT08)".

## Author contributions
P.M.K., F.P.P., E.V., and C.R.B. designed the experiments. F.P.P. performed the mice brain tissue dissection and P.M.K. performed the ribosome complex profiling experiments. K.L. and E.V. conducted the bioinformatic analysis. P.M.K., K.L. and E.V. interpreted the data. All authors discussed the results and contributed to writing the manuscript.

## Funding

## Competing interests
The authors declare no competing interests.
