## [Transparent Peer Review file · Communications Biology]

Transcriptome-wide mapping of small ribosomal subunits elucidates scanning mechanisms of translation initiation in the mammalian brain

Corresponding Author: Dr Eivind Valen

Version 0:

Reviewer comments:

Reviewer #1

(Remarks to the Author)

The authors perform RCP-seq on mouse brain tissue. They introduce a post-lysis UV crosslinking step, which is optimized but not experimentally compared (i.e.) validated to established crosslinking methods. The experimental and bioinformatic methodology is sound. Conclusions are drawn upon the data obtained from three biological replicates of RCP-seq from two mouse brain tissues. No follow up experiments are performed. The study is as such rather descriptive but interesting to the field in its methodological innovation and description of translation in the brain.

Major comments:

-The authors use a post-lysis crosslinking protocol for the fixation of 40S ribosomes on the mRNA. This is different from previous implementations of TCP-seq or RCP-seq. Control experiments should be performed to ensure that post lysis recruitment of 40S ribosomes to the mRNA cannot occur before crosslinking and that this potential process is not the explanation for the detection of "poised" 40S ribosomes.

-The definition of "poised" ribosomes is rather vague and unclear. It is also not unlikely that this is really a new observation.

1) It is important to note that the position of the 3' edge of these 60nt footprint is fully consistent a 40S ribosome on and not upstream of the start codon

2) Queuing ribosome complexes, as indicated in figure 1B would be expect to sediment more deeply in the sucrose gradient (molecular weight of two 40S ribosomes) and would therefore be unlikely to be collected from 40S ribosomal peaks.

3) A previous study (Bohlen et al 2020) had demonstrated that these elongated footprints are enriched for EIF associations by Sel-TCP-seq. This is the opposite of what would be expected from queuing ribosome complexes, in which the ribosome to initiation factor ratio should be the same or lower than single 40S ribosomes.

Therefore, the evidence that the observed -40 position, 60 nt ribosomes footprints represent something different than 40S ribosomes on the start codon, with the initiator tRNA in the p-site base pairing with the start codon is lacking. Re-naming 48S initiation complexes to 'poised' ribosomes will likely be confusing for the field and should be supported by additional data.

For example, the UV crosslinking protocol could be compared to the established in vivo crosslinking method in a readily available cell line such as HEK293 and it should be demonstrated that the UV protocol does not artificially enrich for these 'poised' ribosome footprints. Importantly, an increased presence of initiation 48S complexes on start codons of synaptic mRNAs is still a very interesting observation and does not invalidate the other findings of this manuscript. A mere quantification of the frequency of 40S versus 80S footprints on start codons may reveal the same enrichments observed, without needing to take 40S footprint length into account.

-The authors postulate: "The transition between these two conformations could therefore indicate a regulatory step before or

during start codon recognition concomitant with a change in ribosomal conformational state.”

The authors should consider, discuss and investigate the possibility that these two species are indeed the same complex, with a heterogeneous efficiency of the crosslink in stabilizing the higher molecular weight complex. As such, the two populations would be technical rather than biological in their character. Start codon dwelling time and eIF binding may modulate the crosslinking efficiency.

-Figure 2D: differences in 5'UTR length may be majorly responsible for the reduction in 'poised' ribosomes and 5'UTR occupancy. The authors should demonstrate that this effect is not simply caused by enrichment of short 5'UTRs in the 'low poised ratio' group.

-The authors write: "Interestingly, SSU and 80S footprints showed that neuronally-enriched transcripts are relatively more expressed, scanned and translated compared to glial-enriched transcripts (Figure 4a)." Is this not simply explained by a difference in frequency of neurons versus glia in the sample? If there are more of one type of the cells (neurons) there will be more such transcripts. This refers to the effects observed in figure 4a. The increased TE and SE in figure 4b may still be present, but is a substantially more moderate difference.

Minor Comments:

-References on Page 2 seem to be formatted differently and have a hyperlink format. This should be unified across the manuscript.

-Figure 2C: If data from the cortex samples is going to be presented and not merely referred to in the discussion, the authors should probably present the whole dataset, including QC measures and so on, as well as make the data publicly available after publication. In fact, the authors include a results paragraph towards the end of the results section, this may be more intuitive, if moved closer to the beginning of the results (i.e. before figure 2).

-The authors write: "Interestingly, while monosome-preferring transcripts showed lower TE than polysome-preferring transcripts from the neuropil compartment (Figure 5d), they displayed similar levels of occupancy from scanning ribosomes (Figure 5e) coupled with a higher level of initiation (Figure 5f)."

-The conclusions drawn from figure 5d and 5f seem contradictory. How can there be more initiation if there is less translation? The authors may rephrase this sentence to speculate that dwelling time on the start codon before initiation is potentially elevated in these transcripts? Or maybe this is a typing error, as the data in figure 5f seems to indicate that monosome translated transcripts have a lower initiation rate.

Reviewer #2

(Remarks to the Author)

The manuscript 'Transcriptome-wide mapping of small ribosomal subunits elucidates scanning mechanisms of translation initiation in the mammalian brain' by Kute and colleagues describes the optimization and application of ribosome subunit profiling to brain tissues. The authors demonstrate that their protocol is applicable to the study of different brain regions and showcase how it's able to capture interesting aspects of translation regulation. Although no entirely novel biology is presented, the paper represents a meaningful technical contribution to the field as it could be used to dissect important aspects of translation regulation in the context of neuronal activity, plasticity and pathology. The authors do a good job of framing their work in the context of the relevant literature and drawing interesting comparisons with publicly available datasets. Overall, the manuscript does not suffer from any major conceptual flaw and we feel it's suitable for publication in Communications Biology. However, for the sake of clarity and to improve the manuscript, before publications the authors should address the following important points:

1) In the introduction the authors should provide a succinct explanation of how RCP-seq works, clearly outlining the order of the different steps

2) Supplementary figure 2b is extremely puzzling – how can the trace of the RNaseI treatment show more polysomes than the UV-undigested trace?

3) The schematic of the workflow in figure 1 is not clear. The authors write 'polysome fractionation', but after RNaseI treatment the only species they can recover are 40S, 60S and 80S, so 'SSU and 80S fractionation and collection' seems more appropriate.

4) How can the authors distinguish between poised SSUs and uORFs? Presumably, this is because they collect the fractions prior to sequencing and can discriminate between 40S and 80S reads. However, they should include one line about this to add clarity.

5) The authors claim that neuronal transcripts are translated more efficiently than glial ones. However, from the RNA traces in Figure 4a it seems as though the amount of glial transcripts is substantially lower than that of neuronal transcripts. Do the 80S and SSU traces take this into account? Factoring this in, is it really a fair statement to say that neuronal genes are more efficiently translated? Or are they translated with comparable efficiency relative to transcript amounts?

6) In Fig 4d and 4e the authors provide example traces of neuronal genes, they should also provide example traces for astrocytic transcripts to showcase differences.

7) The metagenome coverage plots in Figure 5C should have the same axis scaling to simplify comparison between compartments.

Reviewer #3

(Remarks to the Author)

Translational control plays a crucial role in the neuronal system and understanding the fundamental mechanisms that regulate mRNA translation in the brain, particularly localized translation, is of significant importance to the field. The development of methods capable of distinguishing initiating ribosome footprints from those of elongating ribosomes represents a major advancement. In this manuscript, Kute et al. have adapted this strategy with notable optimizations, such as the use of UV crosslinking instead of formaldehyde, to map the translational landscape of samples derived from the mouse dentate gyrus and cortex regions. As such, this work has the potential to make a valuable contribution to the field. However, as outlined below, there are substantial concerns regarding the rigour of the methodology and data analyses, and in several instances, the conclusions drawn do not appear to be unequivocally supported by the presented evidence.

Main comments:

1- Although the titles of the three Supplementary Tables are listed in the Supplementary Files section, this reviewer does not appear to have access to the actual tables. In the "Materials and Methods" section, under "Ribosome complex profiling analysis," a hyperlink is provided to a repository where all the processed tables are presumably stored. However, the repository is not organized in alignment with the structure of the manuscript, making it difficult for the reviewer to confidently identify which data corresponds to which table. As a result, this reviewer was unable to properly evaluate these data. However, from what could be surmised from the "Materials and Methods" section, for statistical testing, the authors used "cutoffs of log₂ fold change of 1 or -1 and 0.05 as the significance level." What type of test was used here and does the 0.05 cut-off refer to p-value or adjusted p-value? It is critical that adjusted p-value should be used for this sort of analysis, particularly to assess the differentially translated mRNAs that were used for downstream analyses (e.g. GO analyses). On a related note, t-test is not the most appropriate test for analyses of the type of box-plot data in this study (e.g. Figure 2e, 3b, d, and f). Fisher test would likely be more suitable.

2- How did the authors determine which fractions correspond to the SSU? It is generally assumed that the peak immediately following the floating RNAs corresponds to the 40S subunit. However, given that polysome profiles obtained from brain tissues are often not sharply defined (something that is observed in every study and not just in this manuscript), some form of prior quality control should be conducted to ensure the correct fractions are collected. Perhaps, an analysis of distribution of rRNAs in the fractions? Also, a complementary approach would be to present the ratio of different rRNAs in the libraries. On a related note, can the authors clarify why all the peaks, including those presumed to represent the 40S, 80S, and polysomes, are elevated in RNase I-treated samples compared to the undigested control (Supplementary Figure 1b)? One would typically expect RNase I treatment to result in a significant reduction of polysomes and an increase in the 80S, as well as the 40S and 60S peaks. This figure does not seem to follow the anticipated pattern. Notably, this pattern seems to completely reverse in cortex samples (Supplementary Figure 3b).

3- The distribution of reads from the libraries presented in Supplementary Figures 1c exhibits an unusual pattern. The manuscript states, "The library quality was comparable to previous studies (Archer et al. 2016; Giess et al. 2020) where contaminants were largely from rRNAs and ncRNAs (Supplementary figures 1c)" but does not elaborate further. Can the authors clarify why more than half of the 80S reads and a significant portion of the SSU reads mapped to introns? The parallel RNA-Seq data shown in the same figure indicate that intronic reads constitute only a small fraction of all RNAs and thus cannot account for such high levels of enrichment. Could this pattern suggest that the majority of these reads actually originate from other protein complexes, such as spliceosomes?

4- The manuscript claims "We calculated the translation efficiency (TE, see methods) metric to assess the presence of 80S on these transcripts and observed higher translation efficiency for transcripts with high poisoning ratio as compared to those with low poisoning ratio (Figure 2e)." Do the authors suggest that these two observations are related? i.e., high poisoning ratio correlates with (or leads to) higher translation efficiency?

5- Based on data presented in Figure 3f, the authors propose that "the poised SSUs in those genes containing an active uORF were significantly lower ($p < 0.05$, unpaired t-test, Figure 3e and 3f) suggesting that uORFs may act to slow down or block SSUs on their way to the CDS." If this were the case, there must be an equivalent accumulation of the SSU right before the uORF start codon on these mRNAs. Do the authors see something to that effect?

6- The manuscript states, "Taken together, the neuronal-enriched transcripts contain more SSU reads in their leaders, more 80S coverage in their CDS and are more abundant at the RNA level than glial-enriched transcripts." However, a more plausible interpretation might be that the analyzed samples simply contained a higher proportion of neuronal cells than glial cells, leading to a greater presence of neuronal RNAs. This statement highlights a general tendency in the manuscript to derive potentially equivocal conclusions from descriptive data based on a single type of experiment, even though the data itself is valuable. To strengthen this claim, the authors might consider complementing their findings with immunofluorescence approaches combined with SunSET analysis to provide more direct evidence supporting this observation.

7- Regarding the samples derived from cortex, the manuscript claims that "Footprint length distributions were within the expected range with 20-75 nt for SSU and 28-32 nt for 80S (Supplementary figure 3f)." However, visual inspection of

Supplementary Figure 3f does not show any difference in the read length distribution for SSU vs 80S. This is quite distinct from the similar analysis of samples from DG (Supplementary figure 1e). Can the authors explain the basis of the claim?

Minor comments:

- 1- There is no reference to Supplementary figure 1d in the text.
- 2- What is the significance of visualising the SSU and 80S occupancy profiles for two neuronal transcripts C1ql2 and Prox1 in Figures 4d and 4e. If the purpose is to validate the claim that “neuronally-enriched transcripts are relatively more expressed, scanned and translated compared to glial-enriched transcripts”, it would make more sense to show these in comparison with a few glial specific transcripts.
- 3- It would probably make the manuscript streamline better if the section related to “Mapping of SSU and 80S ribosomes in cortical tissue of adult mouse brain” were presented along with the data from DG and similar analyses where applied to both datasets.

Version 1:

Reviewer comments:

Reviewer #1

(Remarks to the Author)

Thank for your addressing my concerns in a comprehensive and scholarly fashion. Good job on an interesting manuscript that will move the field forward!

Reviewer #2

(Remarks to the Author)

The authors have thoughtfully addressed all comments raised. They clarified methodological details in the introduction, corrected ambiguities in figure labelling and traces, updated schematic representations for accuracy, and provided additional supporting data to enhance clarity and interpretability. These revisions improve the manuscript and strengthen the validity of the authors' conclusions.

In my view, the manuscript is now suitable for publication in Communications Biology without further revision.

Reviewer #3

(Remarks to the Author)

The authors have provided reasonable responses to my queries on the first version of the manuscript and their additional analyses have improved the manuscript. I have no further query.

Reviewer 1

1. The authors perform RCP-seq on mouse brain tissue. They introduce a post-lysis UV crosslinking step, which is optimized but not experimentally compared (i.e.) validated to established crosslinking methods. The experimental and bioinformatic methodology is sound. Conclusions are drawn upon the data obtained from three biological replicates of RCP-seq from two mouse brain tissues. No follow up experiments are performed. The study is as such rather descriptive but interesting to the field in its methodological innovation and description of translation in the brain.

We appreciate that the reviewer finds the study interesting and useful to the field. It is intended as a proof-of-concept for studying translation initiation at the transcriptome-wide level in brain tissue and we therefore do limited functional characterization. To confirm the UV crosslinking protocol, we have now compared the UV and formaldehyde crosslinking methods in HEK293T cells as per the reviewer's suggestion. Below are our responses to the reviewer's constructive and critical comments.

Major comments

2. The authors use a post-lysis crosslinking protocol for the fixation of 40S ribosomes on the mRNA. This is different from previous implementations of TCP-seq or RCP-seq. Control experiments should be performed to ensure that post lysis recruitment of 40S ribosomes to the mRNA cannot occur before crosslinking and that this potential process is not the explanation for the detection of "poised" 40S ribosomes.

We have now performed different crosslinking methods in HEK293T cells to assure that "poised SSUs" are not a result of post-lysis recruitment of 40S ribosomes (supplementary figure 3c-3f). As can be seen, both formaldehyde and UV fixation shows the enrichment of "poised SSUs" which is absent under non-crosslinked conditions (3c).

c-f: Footprint length distribution for the 5' end of SSU fragments in HEK293T cells relative to the TIS, highlighting initiating SSUs (at -12nt) and poised SSUs (-36: -60nt) for different crosslinking condition, for non-crosslinked (**c**), for formaldehyde crosslinked (**d**), for UV crosslinked after lysis (**e**), for in vitro UV crosslinking (**f**)

We also want to point out that 'poised' ribosomes have been shown many times before in multiple studies. Archer et al, Wagner et al, Giess et al, and Bohlen et al. all show similar signatures, although they do not discuss, estimate their distribution across genes or characterize them as extensively as we do. We can therefore conclude that post-lysis recruitment is not the cause of 'poised' 40S ribosomes.

Importantly, in the current study, the lysis and the crosslinking are done in ice-cold conditions so that 40S recruitment and scanning are slowed down or stopped. Thus, we are likely capturing 40S ribosomes in their near-natural states and conformations. In support of this, our signatures over the TIS are highly similar to those of the standard TCP/RCP-seq with no particular enrichment that has not been observed before. While the longer conformation of SSUs is observed in all TCP/RCP-seq studies, SSU queuing, one of the possible conformations of what we refer to as 'poised', has also been suggested during non-AUG translation, slowly elongating ribosomes, and on polyamine stretches (Kearse et al. 2019; Ivanov et al. 2018). For a full overview of instances of queued SSUs in other studies, see our new supplementary table 2.

Supplementary table 2: overview of descriptions of queued SSU

Reference	Model system	Localization	Suggested mechanism?	Detection method
(Ivanov et al. 2018)	HEK293T cells, reporter assays	SSU queuing on uORF, upstream of paused ribosome	high polyamines, elongation pausing (over PPW motif) during translation of the uORF	Model
(Kearse et al. 2019)	HeLa cells, reporter assays	SSU queuing on ORF, upstream of paused ribosome	non-AUG start codons, slowed elongating ribosomes	Model
(Archer et al. 2016)	Yeast	SSU queuing on leaders with 5' end -30nt rel to TIS	local structural or sequence features of mRNAs	TCP-seq
(Wagner et al. 2020)	Yeast and HEK293T	In yeast, SSU queuing on leaders -30nt rel to TIS, for HEK293T SSU queuing on leaders -60 to -30nt rel to TIS		Sel-TCP seq and TCP-seq
(Bohlen et al. 2020)	NIH 3T3	SSU queuing on leaders -120nt to -60nt rel to TIS	80S ribosomes stalled at >12 codons downstream of TIS	Harringtonine treatment and TCP-seq

3. The definition of "poised" ribosomes is rather vague and unclear. It is also not unlikely that this is really a new observation.

The enrichment of long SSU fragments upstream of the TIS is not a new observation. However, this study looks more closely at the phenomenon of SSU queuing, and particularly at those that are in the -40 region upstream of the TIS, and finds these to be enriched in particular functionally-related transcripts. These longer fragments are typically attributed to the longer conformations of SSUs with initiation factors (~60 nt), but can also be compatible with SSUs queuing (2x~30nt). In both cases the SSUs are in a poised conformation, waiting to transition to elongation. Therefore, the term “poised” SSU encompasses two possible conformations presented in the literature: 1) one SSU with initiation factors or 2) two SSUs queuing. We have clarified this in the text (page 9, lines 4-8), figure 3b, and figure legends.

b

$$\text{SSU poised ratio} = \frac{\text{Fragments 60-65nt at -46:-36}}{\text{Fragments 25-35nt at -14:-12}}$$

b: Schematic of different conformations of SSUs near the TIS of transcripts. Formulae for calculating SSU poised ratio.

While SSU queuing has been described in earlier studies (summarised in the table above), in this study, we estimate the amount of SSU “poising” relative to the initiating SSUs by computing a novel metric. This is a ratio of two conformations originally described in the Archer et al paper and captures the relative amounts of these. To clarify we have now added schematics on the definition of poised SSUs (Figures 3b). Additionally, we direct the reviewer to the metric of the SSU poised ratio in Figure 3b.

SSU poised ratio: Long SSU (60-65nt) at -46: -36 relative to TIS/ Short SSU (25-35nt) at -14: -12 relative to TIS

An important point is also that regardless of what conformation these SSUs occupy (e.g. if we exclusively referred to these as ‘48S initiation complexes’), the ‘poised ratio’ and enrichment would still describe a difference between conformations in terms of translation dynamics during initiation. But we would rather not exclude potentially valid interpretations by claiming certainty about only one of these.

4. It is important to note that the position of the 3’ edge of these 60nt footprint is fully consistent a 40S ribosome on and not upstream of the start codon

Yes, we agree and this is outlined in Figure 3b (earlier figure 2b) as one of two possibilities for a poised SSU. The following figures for DG and cortex show the two SSU populations; the short SSU with 5’ ends at -14: -12 and the long SSU with 5’ ends at -46: -36 and with overlapping 3’ ends at 10: 26. These figures are also added to supplementary figure 3.

a and **b**: Frequency distribution of SSU footprints based on their 5' end position on the leaders for the DG (**a**) and for the cerebral cortex (**b**). Numbers on the plots indicate nucleotide positions on the leaders relative to the TIS/start codon.

5. Queueing ribosome complexes, as indicated in figure 1B would be expected to sediment more deeply in the sucrose gradient (molecular weight of two 40S ribosomes) and would therefore be unlikely to be collected from 40S ribosomal peaks.

As suggested by Archer et al. 2016 it is likely that in sucrose gradients (10-50%), two 40Ss co-sediment with a single 40Ss as a broader peak (see Archer et al. 2016, fig 1b inset). As they write for conformations in yeast:

"Footprints ending at +16 and +24 nt increasingly developed a second population with footprint 5' ends around -30 nt, possibly the result of additional scanning SSU 'queueing' and further extending protection upstream by ~19 nt, due to cap recruitment and scanning outpacing start codon clearance (Fig. 4b, middle and bottom), although other explanations are also plausible. The shape of the SSU-related sedimentation peak (Fig. 1b) is also consistent with such complex heterogeneity."

Our interpretations are also based on the footprint lengths we obtained and the observations and interpretations made by other SSU studies in yeast (Archer et al. 2016; Wagner et al. 2020) and zebrafish embryos (Giess et al. 2020).

6. A previous study (Bohlen et al 2020) had demonstrated that these elongated footprints are enriched for eIF associations by Sel-TCP-seq. This is the opposite of what would be expected from queueing ribosome complexes, in which the ribosome to initiation factor ratio should be the same or lower than single 40S ribosomes.

Based purely on read lengths and sedimentation we cannot distinguish between IFs associated with single 40Ss and queued 40Ss. The elongated footprints from Bohlen correspond to conformation #2 below and are indeed enriched for IFs. However, SSU queueing has also been observed and discussed multiple times, including in Bohlen et al (although more upstream than what we observe). They have shown this using harringtonine treatment in a mouse cell line (NH3T3T) leading to SSU queueing, possibly due to the recruitment of multiple SSUs and slower initiation. Since we cannot differentiate queueing from initiating ribosomes associated with IFs, we collectively refer to these two complexes as poised.

Cartoons depicting different SSU conformations and initiation stages based on footprint lengths and 5' and 3' end extensions. Similarly depicted in figure 3b.

7. Therefore, the evidence that the observed -40 position, 60 nt ribosomes footprints represent something different than 40S ribosomes on the start codon, with the initiator tRNA in the p-site base pairing with the start codon is lacking. Re-naming 48S initiation complexes to 'poised' ribosomes will likely be confusing for the field and should be supported by additional data.

As illustrated above, poised SSU conformations refer to conformation 2 and 3. In 2, a single PIC with trailing IFs extends up to -42 nt. In 3, two adjacent SSUs extend up to 42 nt. It is therefore not intended to replace "48S initiation complex", but to be a more general term.

8. For example, the UV crosslinking protocol could be compared to the established in vivo crosslinking method in a readily available cell line such as HEK293T and it should be demonstrated that the UV protocol does not artificially enrich for these 'poised' ribosome footprints.

We thank the reviewer for suggesting these experiments with HEK293T. We have used formaldehyde to crosslink SSU complexes in HEK293T cells. As seen below, crosslinking of HEK293T with formaldehyde (0.3%) shows a population of SSUs (at -36 to -60 and 60-80 nt long) conformation in addition to a population at -14 to -12. It is thus unlikely that this conformation comes from UV crosslinking specifically. This is part of supplementary figure 3.

c-f: Footprint length distribution for the 5' end of SSU fragments in HEK293T cells relative to the TIS, highlighting initiating SSUs (at -12nt) and poised SSUs (-36: -60nt) for different crosslinking condition, for non-crosslinked (**c**), for formaldehyde crosslinked (**d**), for UV crosslinked after lysis (**e**), for in vitro UV crosslinking (**f**)

Importantly, an increased presence of initiation 48S complexes on start codons of synaptic mRNAs is still a very interesting observation and does not invalidate the other findings of this manuscript. A mere quantification of the frequency of 40S versus 80S footprints on start codons may reveal the same enrichments observed, without needing to take 40S footprint length into account.

When we look at the quantification of 40S versus 80S footprints on the start codons, it correlates poorly with the SSU poised ratio, as seen below. Thus, the two do not seem to represent the same metric nor the same translation event (see below).

Figure for reviewers only.

9. The authors postulate: “The transition between these two conformations could therefore indicate a regulatory step before or during start codon recognition concomitant with a change in ribosomal conformational state.”

The authors should consider, discuss and investigate the possibility that these two species are indeed the same complex, with a heterogeneous efficiency of the crosslink in stabilizing the higher molecular weight complex. As such, the two populations would be technical rather than biological in their character. Start codon dwelling time and eIF binding may modulate the crosslinking efficiency.

We thank the reviewer for his concerns about the existence of the two conformations that might come from the ambiguity of our definitions. These short and poised conformations have been described before by Archer et al., 2016. Furthermore, to clarify these conformations we added schematics on the definition of short and poised SSUs in Figure 3b. We also performed formaldehyde crosslinking in HEK293T cells following the protocol of Wagner et al (Wagner et al. 2020) and observed a similar enrichment of SSUs with 5' ends around -50nt and 3' ends around 25 relative to the TIS (Supplementary figures 3d), indicating the similar poised SSU conformations observed in the brain tissues.

10. Figure 2D: differences in 5'UTR length may be majorly responsible for the reduction in 'poised' ribosomes and 5'UTR occupancy. The authors should demonstrate that this effect is not simply caused by enrichment of short 5'UTRs in the 'low poised ratio' group.

We thank the reviewer for suggesting new control analyses of our data. We have now visualized 'poised' as a function of 5' UTR length which can be seen in the figure below. 5'UTR length does not appear to correlate with the poising ratio (see below and supplementary figure 4).

Supplementary figure 4: Effect of features of leaders on SSU poising (related to figure 4)

a: Log2 of SSU poised ratio vs. the length of the 5' UTRs for transcripts from DG.

11. The authors write: “Interestingly, SSU and 80S footprints showed that neuronally-enriched transcripts are relatively more expressed, scanned and translated compared to glial-enriched transcripts (Figure 4a).” Is this not simply explained by a difference in frequency of neurons versus glia in the sample? If there are more of one type of the cells (neurons) there will be more such transcripts. This refers to the effects observed in figure 4a. The increased TE and SE in figure 4b may still be present, but is a substantially more moderate difference.

Yes, we agree with the reviewer that the expression of these mRNAs likely reflect the abundance of neurons and glia in the tissue. Indeed a previous study showed that in the adult mouse DG, mature neurons represent 87% of the total number of cells compared to astrocytes, microglia, and oligodendrocytes (Rieskamp et al. 2022). It is now stated in the main text of the manuscript (page 15, lines 13-15: “Neuronally-enriched transcripts show more coverage of SSU, 80S and RNA as compared to glial-enriched transcripts which may be reflective of the cell-type abundances in the tissue (Figure 5a)”).

However, the metrics TE and SE are independent of the cell type abundance as they are normalised to RNA. We have now removed the term “expressed” as this is misleading.

Minor Comments

12. References on Page 2 seem to be formatted differently and have a hyperlink format. This should be unified across the manuscript.

We have fixed the references, and we apologize for the formatting error.

13. Figure 2C: If data from the cortex samples is going to be presented and not merely referred to in the discussion, the authors should probably present the whole dataset, including QC measures and so on, as well as make the data publicly available after publication. In fact, the authors include a results paragraph towards the end of the results section, this may be more intuitive, if moved closer to the beginning of the results (i.e. before figure 2).

Yes, we have moved the cortex section closer to the beginning of the results (it is Figure 2).

14. The authors write: “Interestingly, while monosome-preferring transcripts showed lower TE than polysome-preferring transcripts from the neuropil compartment (Figure 5d), they displayed similar levels of occupancy from scanning ribosomes (Figure 5e) coupled with a higher level of initiation (Figure 5f).”

The conclusions drawn from figure 5d and 5f seem contradictory. How can there be more initiation if there is less translation? The authors may rephrase this sentence to speculate that dwelling time on the start codon before initiation is potentially elevated in these transcripts? Or maybe this is a typing error, as the data in figure 5f seems to indicate that monosome translated transcripts have a lower initiation rate.

We thank the reviewer for carefully reading our study and identifying that mistake. Indeed, monosomally translated mRNAs show lower initiation rates than polysomally translated mRNA and it has been fixed in the manuscript (page 17, line 16).

Reviewer 2

15. The manuscript 'Transcriptome-wide mapping of small ribosomal subunits elucidates scanning mechanisms of translation initiation in the mammalian brain' by Kute and colleagues describes the optimization and application of ribosome subunit profiling to brain tissues. The authors demonstrate that their protocol is applicable to the study of different brain regions and showcase how it's able to capture interesting aspects of translation regulation. Although no entirely novel biology is presented, the paper represents a meaningful technical contribution to the field as it could be used to dissect important aspects of translation regulation in the context of neuronal activity, plasticity and pathology. The authors do a good job of framing their work in the context of the relevant literature and drawing interesting comparisons with publicly available datasets. Overall, the manuscript does not suffer from any major conceptual flaw and we feel it's suitable for publication in Communications Biology. However, for the sake of clarity and to improve the manuscript, before publications the authors should address the following important points

We thank the reviewer for finding the manuscript suitable for publication. We have addressed the comments proposed by the reviewer and improved the manuscript accordingly.

16. In the introduction the authors should provide a succinct explanation of how RCP-seq works, clearly outlining the order of the different steps

This has now been addressed in the manuscript and highlighted in blue in the introduction section (page 3, lines 6-12).

17. Supplementary Figure 2b is extremely puzzling – how can the trace of the RNaseI treatment show more polysomes than the UV-undigested trace?

We agree with the reviewer that RNase digestion should lead to a reduction in polysomes. Below are the original traces that were smoothed (the noise in the beginning and the small spike in the red trace were removed) for the manuscript. Differences in the sucrose gradients can cause the peaks to misalign or go higher than the control (the drop at the end in the two traces shows the misalignment). The blue trace is an undigested sample and the red trace is after RNase 1 treatment. We have now used these traces in the manuscript to avoid confusion.

Original A254 trace for supplementary figure 1b.

18. The schematic of the workflow in figure 1 is not clear. The authors write 'polysome fractionation', but after RNaseI treatment the only species they can recover are 40S, 60S and 80S, so 'SSU and 80S fractionation and collection' seems more appropriate.

We appreciate the input to improve the accuracy of the wording. This has been addressed and Figure 1a has been updated.

a

Figure 1: Ribosome complex profiling (RCP-seq) captures small ribosomal subunits (SSU) and elongating ribosomes (80S) from the mouse DG tissue.

a: Steps for ribosomal complex profiling using UV crosslinking for the fixation of the SSU and 80S in the DG.

19. How can the authors distinguish between poised SSUs and uORFs? Presumably, this is because they collect the fractions prior to sequencing and can discriminate between 40S and 80S reads. However, they should include one line about this to add clarity.

We are unsure about how to interpret this concern. If the reviewer is referring to how we distinguish between poised SSUs and 80S, the answer is: yes, the respective fractions are collected based on the UV trace post-digestion and subjected to library preparation as described in the material/method section. As per the reviewer's previous suggestion, we have explained RCP-seq in the introduction (page 3, lines 6-12). We have also added a total RNA bioanalyzer profile for fractions from DG and cortical polysomes highlighting the absence of 28S rRNA in the fractions containing 18S rRNA demonstrating that they are 40S fractions (Supplementary figures 1c, 1d, 2a, and 2b). Furthermore, a clarified definition of poised SSUs is now illustrated in Figure 3b.

c

d

Supplementary Figure 1: Quality assessment of the UV-crosslinking method for RCP-seq (Related to Figure 1)

c: Amount of crosslinking for polysomes for three different cross-linking methods: formaldehyde, UV or non-crosslinked DG samples.

d: Polysome profiles of UV-crosslinked, with or without RNase I digestion to denote an increase in 80S for digested samples for DG

Supplementary Figure 2: UV-crosslinking method for RCP-seq in the cortical tissue

a: A UV trace of cortical polysomes treated only with cycloheximide (no crosslinking) from a 10-50% sucrose gradient. 20 fractions were collected out of which 15 fractions are shown.

b: Bioanalyzer image of total RNA isolated from fractions from the above profile. From this image, fractions 3 and 4 may contain 40S.

b: Schematic of different conformations of SSUs near the TIS of transcripts. Formulae for calculating SSU poised ratio.

20. The authors claim that neuronal transcripts are translated more efficiently than glial ones. However, from the RNA traces in Figure 4a it seems as though the amount of glial transcripts

is substantially lower than that of neuronal transcripts. Do the 80S and SSU traces take this into account? Factoring this in, is it really a fair statement to say that neuronal genes are more efficiently translated? Or are they translated with comparable efficiency relative to transcript amounts?

Figure 5a (previously Figure 4a) shows how much neuron or glia-enriched transcripts are scanned and translated, which may reflect the contribution of cell type-specific transcripts to the total RNA pool of the tissue. Figure 5b and 5c shows analysis of translational efficiency and scanning efficiencies (i.e. normalized by RNA abundance). This takes into account the abundance of neuron- and glia-specific transcripts by normalising 80S/SSU reads with the respective RNA. In light of this, it is fair to say that neuronal-enriched transcripts are translated more efficiently than glial-enriched transcripts.

21. In Fig 4d and 4e the authors provide example traces of neuronal genes, they should also provide example traces for astrocytic transcripts to showcase differences.

We thank the reviewer for the suggestion. To exemplify differences in SSU and 80S occupancy, we have now added plots for glial-enriched transcripts *Tgfb2* and *Fth1* (Figures 5f and 5g).

f) *Tgfb2* (Transforming growth factor beta 2), **g)** *Fth1* (Ferritin heavy chain 1). Numbers indicate FPKM values. Dotted lines indicate TIS (red) and TTS (black).

22. The metagene coverage plots in Figure 5C should have the same axis scaling to simplify comparison between compartments.

We have now adjusted the axis for Figure 6b and 6c (previously 5b and 5c).

Figure 6: Translation landscape of neuropil and soma-enriched transcripts

b: Metagene coverage for transcripts enriched in the soma and preferring monosomal or polysomal translation.

c: Metagene coverage for transcripts enriched in the neuropil and preferring monosomal or polysomal translation.

Reviewer 3

23. Translational control plays a crucial role in the neuronal system and understanding the fundamental mechanisms that regulate mRNA translation in the brain, particularly localized translation, is of significant importance to the field. The development of methods capable of distinguishing initiating ribosome footprints from those of elongating ribosomes represents a major advancement. In this manuscript, Kute et al. have adapted this strategy with notable optimizations, such as the use of UV crosslinking instead of formaldehyde, to map the translational landscape of samples derived from the mouse dentate gyrus and cortex regions. As such, this work has the potential to make a valuable contribution to the field. However, as outlined below, there are substantial concerns regarding the rigour of the methodology and data analyses, and in several instances, the conclusions drawn do not appear to be unequivocally supported by the presented evidence.

We thank the reviewer for taking the time to review the manuscript and giving their critical comments and concerns. Please find below our responses to the comments and suggestions.

Major comments

24. Although the titles of the three Supplementary Tables are listed in the Supplementary Files section, this reviewer does not appear to have access to the actual tables.

We apologize for the omission. We have now uploaded the supplementary tables.

25. In the “Materials and Methods” section, under “Ribosome complex profiling analysis,” a hyperlink is provided to a repository where all the processed tables are presumably stored. However, the repository is not organized in alignment with the structure of the manuscript, making it difficult for the reviewer to confidently identify which data corresponds to which table. As a result, this reviewer was unable to properly evaluate these data.

We apologize for this. While it is not straightforward to rearrange the ordering of a GitHub repository we have now added a list that links all the figures and tables to their respective files.

https://git.app.uib.no/valenlab/brain_dg_cortex_rcp/-/blob/master/Map%20of%20figures%20to%20their%20paths.xlsx?ref_type=heads

26. However, from what could be surmised from the “Materials and Methods” section, for statistical testing, the authors used “cutoffs of log₂ fold change of 1 or -1 and 0.05 as the significance level.” What type of test was used here and does the 0.05 cut-off refer to p-value or adjusted p-value? It is critical that adjusted p-value should be used for this sort of analysis, particularly to assess the differentially translated mRNAs that were used for downstream analyses (e.g. GO analyses). On a related note, t-test is not the most appropriate test for analyses of the type of box-plot data in this study (e.g. Figure 2e, 3b, d, and f). Fisher test would likely be more suitable.

We have consistently used adjusted p-values when multiple testing correction is warranted. This is a textual omission that has now been corrected. Regarding the type of test for box plots, since we are comparing whether the mean of two distributions differs significantly from each other we would argue that a t-test is the most appropriate test under a normality assumption. Fisher's exact test would require us to create categorical/ordinal variables from continuous data which would rely on ad hoc thresholds. We hope the reviewer agrees with us on this point.

27. How did the authors determine which fractions correspond to the SSU? It is generally assumed that the peak immediately following the floating RNAs corresponds to the 40S subunit. However, given that polysome profiles obtained from brain tissues are often not sharply defined (something that is observed in every study and not just in this manuscript), some form of prior quality control should be conducted to ensure the correct fractions are collected. Perhaps, an analysis of distribution of rRNAs in the fractions? Also, a complementary approach would be to present the ratio of different rRNAs in the libraries.

We do indeed use analysis of rRNA distribution across fractions to isolate the 40S and we thank the reviewer for pointing out that this should be included. We have now added a total RNA bio analyser for the polysome fractions to show the distribution of 18S and 28S rRNA clearly separating the 40S, 60S, and 80S fractions (Supplementary figures 1c, 1d, 2a, and 2b). This rRNA distribution guided us in isolating the 40S fraction for the relevant experiments.

c: A UV trace of DG polysomes treated only with cycloheximide (no crosslinking) from a 10-15% sucrose gradient. 20 fractions were collected.

d: Bioanalyzer image of total RNA isolated from fractions from the above profile. From this image, fractions 5 and 6 contain 40S.

Supplementary Figure 2: UV-crosslinking method for RCP-seq in the cortical tissue

a: A UV trace of cortical polysomes treated only with cycloheximide (no crosslinking) from a 10-50% sucrose gradient. 20 fractions were collected out of which 15 fractions are shown.

b: Bioanalyzer image of total RNA isolated from fractions from the above profile. From this image, fractions 3 and 4 may contain 40S.

28. On a related note, can the authors clarify why all the peaks, including those presumed to represent the 40S, 80S, and polysomes, are elevated in RNase I-treated samples compared to the undigested control (Supplementary Figure 1b)? One would typically expect RNase I treatment to result in a significant reduction of polysomes and an increase in the 80S, as well as the 40S and 60S peaks. This figure does not seem to follow the anticipated pattern. Notably, this pattern seems to completely reverse in cortex samples (Supplementary Figure 3b).

We agree with the reviewer that RNase digestion should lead to a reduction in polysomes. Below are the original traces that were smoothed (the noise in the beginning and the small spike in the red trace were removed) for the manuscript. Differences in the sucrose gradients can cause the peaks to misalign or go higher than the control (the drop at the end in the two traces shows the misalignment). The blue trace is an undigested sample and the red trace is after RNase 1 treatment. We have now used these traces in the manuscript to avoid confusion.

Original A254 trace for supplementary figure 1b

Smoothed A254 trace (supplementary figure 1b)

For the cortical samples, the labels were unfortunately swapped while making the figures. Find below the original and the smoothed graphs of the cortical polysome profiles

Original A254 trace for supplementary figure 2d

Smoothed A254 trace (Supplementary figure 2d)

29. The distribution of reads from the libraries presented in Supplementary Figures 1c exhibits an unusual pattern. The manuscript states, "The library quality was comparable to previous studies (Archer et al. 2016; Giess et al. 2020) where contaminants were largely from rRNAs and ncRNAs (Supplementary figures 1c)" but does not elaborate further. Can the authors clarify why more than half of the 80S reads and a significant portion of the SSU reads mapped to introns? The parallel RNA-Seq data shown in the same figure indicate that intronic reads constitute only a small fraction of all RNAs and thus cannot account for such high levels of enrichment. Could this pattern suggest that the majority of these reads actually originate from other protein complexes, such as spliceosomes?

More careful reclassification of the reads revealed that the reads mapping to introns were arising from reads that overlapped with either rRNA or other categories. Below is the final mapping of the libraries. We have updated the supplementary figures (Suppl figure 1e and Suppl figure 2e)

Mapping for both DG and cortical tissues

30. The manuscript claims “We calculated the translation efficiency (TE, see methods) metric to assess the presence of 80S on these transcripts and observed higher translation efficiency for transcripts with high poisoning ratio as compared to those with low poisoning ratio (Figure 2e).” Do the authors suggest that these two observations are related? i.e., high poisoning ratio correlates with (or leads to) higher translation efficiency?

We simply observe that SSU poisoning is associated with high translation efficiencies. We have elaborated with some speculation on this in the discussion section (page 20, lines 10-14).

“In the mouse brain, SSU poisoning could potentially be a mechanism for transcripts with slow elongation or poor start codon recognition rates and could allow for fast and abundant translation in response to stimuli, such as during neuronal activity-induced bursts of protein synthesis. Supporting the later hypothesis, we find that genes with high SSU poisoning ratio show higher translation efficiency as compared to those with lower SSU poisoning.”

31. Based on data presented in Figure 3f, the authors propose that “the poised SSUs in those genes containing an active uORF were significantly lower ($p < 0.05$, unpaired t-test, Figure 3e and 3f) suggesting that uORFs may act to slow down or block SSUs on their way to the CDS.” If this were the case, there must be an equivalent accumulation of the SSU right before the uORF start codon on these mRNAs. Do the authors see something to that effect?

We have now discussed this possibility in the main text (page 13, lines 10-14) and see Supplementary figure 4b. We think that uORF initiation is less stringent and more stochastic and we therefore do not expect an accumulation to the same extent here. Rather the accumulation will then be spread out over, potentially multiple, uORF start codons and the CDS start codon.

Supplementary figure 4: Effect of leaders on SSU poisoning (related to figure 4)

b: Footprint length distribution for the 5' end SSU fragments relative to TIS of active uORFs in the DG tissue, highlighting initiating SSUs (at -12nt) and poised SSUs (-46: -42nt).

32. The manuscript states, “Taken together, the neuronal-enriched transcripts contain more SSU reads in their leaders, more 80S coverage in their CDS and are more abundant at the RNA level than glial-enriched transcripts.” However, a more plausible interpretation might be that the analyzed samples simply contained a higher proportion of neuronal cells than glial cells, leading to a greater presence of neuronal RNAs. This statement highlights a general tendency in the manuscript to derive potentially equivocal conclusions from descriptive data based on a single type of experiment, even though the data itself is valuable. To strengthen this claim, the authors might consider complementing their findings with immunofluorescence approaches combined with SunSET analysis to provide more direct evidence supporting this observation.

We agree with the reviewer that the expression of these mRNAs likely reflect the abundance of neurons and glia in the tissue. Indeed a previous study showed that in the adult mouse DG, mature neurons represent 87% of the total number of cells compared to astrocytes, microglia, and oligodendrocytes (Rieskamp et al. 2022). It is now stated in the main text of the manuscript (page 15, lines 13-15: “Neuronally-enriched transcripts show more coverage of SSU, 80S and RNA as compared to glial-enriched transcripts which may be reflective of the cell-type abundances in the tissue (Figure 5a)”). However, the metrics TE and SE are independent of the cell type abundance as they are normalised to RNA. We have now removed the term “expressed” as this is misleading.

We believe that nascent-peptide labelling and immunofluorescence assays would be powerful approaches for comparing neuron vs glia translational output, but we consider this beyond the scope of the current work which focuses specifically on scanning and initiation and the first application of RCP-seq to brain tissue. A fully valid comparison by SunSET would require local intrahippocampal infusion of puromycin and immunostaining analysis of tissue sections, in which visual separation of densely packed neuronal (dendritic) and glia processes poses a formidable challenge. We hope our findings will inspire future studies on small ribosomal subunit analysis in mammalian brain using cell type-specific labeling and sorting of subcellular compartments such as synaptosomes.

33. Regarding the samples derived from cortex, the manuscript claims that “Footprint length distributions were within the expected range with 20-75 nt for SSU and 28-32 nt for 80S (Supplementary figure 3f).” However, visual inspection of Supplementary Figure 3f does not show any difference in the read length distribution for SSU vs 80S. This is quite distinct from the similar analysis of samples from DG (Supplementary figure 1e). Can the authors explain the basis of the claim?

For the cortex, there is still an enrichment of longer fragments versus shorter fragments (~30nt) in SSU (about 6-fold) versus 80S (about 4-fold). We agree that this enrichment is much less pronounced than in DG. The reason for this is unclear to us. Longer footprints (up to 60 nt) in 80S libraries have previously been attributed to the presence of initiation factors or a trailing SSU unit as has been discussed in a previous TCP-seq study (Wagner et al. 2020) or ribosomal interactions with other RNAs such as lncRNAs (Ruiz-Orera and Albà 2019). For clarity, we have rewritten this description in the main text (Page 4-5, lines 25-26, 1-2).

Minor comments

34. There is no reference to Supplementary figure 1d in the text.

We thank the reviewer for pointing this out, we have now cited this figure in the text (page 5, lines 11-12).

35. What is the significance of visualising the SSU and 80S occupancy profiles for two neuronal transcripts C1ql2 and Prox1 in Figures 4d and 4e. If the purpose is to validate the claim that “neuronally-enriched transcripts are relatively more expressed, scanned and translated compared to glial-enriched transcripts”, it would make more sense to show these in comparison with a few glial specific transcripts.

We thank the reviewer for the suggestion, glial transcripts are now added in supplementary figures 5f and 5g (now figure 6b and 6c).

Figure 6: Translation landscape of neuropil and soma-enriched transcripts

b: Metagene coverage for transcripts enriched in the soma and preferring monosomal or polysomal translation.

c: Metagene coverage for transcripts enriched in the neuropil and preferring monosomal or polysomal translation.

36. It would probably make the manuscript streamline better if the section related to “Mapping of SSU and 80S ribosomes in cortical tissue of adult mouse brain” were presented along with the data from DG and similar analyses were applied to both datasets.

We thank the reviewer for the suggestion. We have rearranged the cortex section and moved it up as Figure 2

References

- Archer, Stuart K., Nikolay E. Shirokikh, Traude H. Beilharz, and Thomas Preiss. 2016. “Dynamics of Ribosome Scanning and Recycling Revealed by Translation Complex Profiling.” *Nature* 535 (7613): 570–74.
- Bohlen, Jonathan, Kai Fenzl, Günter Kramer, Bernd Bukau, and Aurelio A. Teleman. 2020. “Selective 40S Footprinting Reveals Cap-Tethered Ribosome Scanning in Human Cells.” *Molecular Cell* 79 (4): 561–74.e5.
- Giess, Adam, Yamila N. Torres Cleuren, Håkon Tjeldnes, Maximilian Krause, Teshome Tilahun Bizuayehu, Senna Hiensch, Aniekan Okon, Carston R. Wagner, and Eivind Valen. 2020. “Profiling of Small Ribosomal Subunits Reveals Modes and Regulation of Translation Initiation.” *Cell Reports* 31 (3): 107534.
- Ivanov, Ivaylo P., Byung-Sik Shin, Gary Loughran, Ioanna Tzani, Sara K. Young-Baird, Chune Cao, John F. Atkins, and Thomas E. Dever. 2018. “Polyamine Control of Translation Elongation Regulates Start Site Selection on Antizyme Inhibitor mRNA via Ribosome Queuing.” *Molecular Cell* 70 (2): 254–64.e6.
- Kearse, Michael G., Daniel H. Goldman, Jiou Choi, Chike Nwaezeapu, Dongming Liang, Katelyn M. Green, Aaron C. Goldstrohm, Peter K. Todd, Rachel Green, and Jeremy E. Wilusz. 2019. “Ribosome Queuing Enables Non-AUG Translation to Be Resistant to Multiple Protein Synthesis Inhibitors.” *Genes & Development* 33 (13-14): 871–85.
- Rieskamp, Joshua D., Patricia Sarchet, Bryon M. Smith, and Elizabeth D. Kirby. 2022. “Estimation of the Density of Neural, Glial, and Endothelial Lineage Cells in the Adult Mouse Dentate Gyrus.” *Neural Regeneration Research* 17 (6): 1286–92.
- Ruiz-Orera, Jorge, and M. Mar Albà. 2019. “Conserved Regions in Long Non-Coding RNAs Contain Abundant Translation and Protein-RNA Interaction Signatures.” *NAR Genomics and Bioinformatics* 1 (1): e2.
- Wagner, Susan, Anna Herrmannová, Vladislava Hronová, Stanislava Gunišová, Neelam D. Sen, Ross D. Hannan, Alan G. Hinnebusch, Nikolay E. Shirokikh, Thomas Preiss, and Leoš Shivaya Valášek. 2020. “Selective Translation Complex Profiling Reveals Staged Initiation and Co-Translational Assembly of Initiation Factor Complexes.” *Molecular Cell* 79 (4): 546–60.e7.